



# Compositional changes of present-day transatlantic Saharan dust deposition

Laura F. Korte[1], Geert-Jan A. Brummer[1,2], Michèlle van der Does[1], Catarina V. Guerreiro[3],
Rick Hennekam[1], Johannes A. van Hateren[1,2], Dirk Jong[1], Chris I. Munday[1], Stefan
Schouten[4], Jan-Berend W. Stuut[1,5]

[1] NIOZ – Royal Netherlands Institute for Sea Research, Department of Ocean Systems, and Utrecht University,
Texel, the Netherlands
[2] Faculty of Earth and Life Sciences, Vrije Universiteit Amsterdam, the Netherlands
[3] University of Bremen, Faculty of Earth Sciences, Bremen, Germany
[4] NIOZ – Royal Netherlands Institute for Sea Research, Department of Marine Microbiology and
Biogeochemistry, and Utrecht University, Texel, the Netherlands
[5] MARUM – Center for Marine Environmental Sciences, University of Bremen, Germany

*Correspondence to*: Laura F. Korte (laura.korte@nioz.nl)

**Abstract.** Massive amounts of Saharan dust are blown from the African coast across the Atlantic Ocean towards
the Americas each year. This dust has, depending on its chemistry, direct and indirect effects on global climate
including reflection and absorption of solar radiation as well as transport and deposition of nutrients and metals
fertilizing both ocean and land. To determine the temporal and spatial variability of Saharan dust transport and
deposition and their marine environmental effects across the equatorial North Atlantic Ocean, we have set up a
monitoring experiment using deep-ocean sediment traps as well as land-based dust collectors. The sediment traps
were deployed at five ocean sites along a transatlantic transect between northwest Africa and the Caribbean along
12° N, in a down-wind extension of the land-based dust collectors placed at 19° N on the Mauritanian coast in
Iwik. In this paper, we lay out the setup of the monitoring experiment and present the particle fluxes from sediment
trap sampling over 24 continuous and synchronised intervals from October 2012 through to November 2013. We
establish the temporal distribution of the particle fluxes deposited in the Atlantic and compare chemical
compositions with the land-based dust collectors propagating to the down-wind sediment trap sites, and with
satellite observations of Saharan dust outbreaks.

First-year results show that the total mass fluxes in the ocean are highest at the sampling sites in the east and west,
closest to the African continent and the Caribbean, respectively. Element ratios reveal that the lithogenic particles
deposited nearest to Africa are most similar in composition to the Saharan dust collected in Iwik. Down-wind
increasing Al, Fe and K contents suggest a downwind change in the mineralogical composition of Saharan dust
and indicate an increasing contribution of clay minerals towards the west. In the westernmost Atlantic, admixture
of re-suspended clay-sized sediments advected towards the deep sediment trap cannot be excluded. Seasonality is
most prominent near both continents but generally weak, with mass fluxes dominated by calcium carbonate and
clear seasonal maxima of biogenic silica towards the west. The monitoring experiment is now extended with
autonomous dust sampling buoys for better quantification Saharan dust transport and deposition from source to
sink and its impact on fertilization and carbon export to the deep ocean.

**Keywords** Saharan dust; North Atlantic Ocean; sediment traps; particle fluxes; XRF element ratios, transatlantic





## 1 Introduction

Latest estimates of transatlantic Saharan-dust transport and deposition based on 3-D satellite imagery indicate that on a yearly basis (2007 - 2013) an average amount of 182 Tg dust is blown from the northwest African coast at 15°W and transported westward towards the Americas (Yu et al., 2015a). Of this dust, about 132 Tg a$^{-1}$ reaches

35°W and 43 Tg a$^{-1}$ reaches 75°W, meaning that around 50 Tg a$^{-1}$ of dust is deposited into the eastern equatorial North Atlantic Ocean between 10°S and 30°N and 140 Tg a$^{-1}$ of dust is deposited into the equatorial North Atlantic Ocean and the Caribbean Sea as well as onto parts of the Amazon rainforest. Due to the dust's impact on global climate (e. g. Goudie and Middleton, 2001;Griffin et al., 2001;Jickells et al., 2005;Maher et al., 2010;Mahowald et al., 2014), Saharan dust has been examined extensively on either side of the equatorial North Atlantic Ocean.

At, and in vicinity to Saharan dust sources, remote-sensing tools were used for example by Schepanski et al. (2007) and Evan et al. (2015) to map Saharan dust source activation and dust emission in North Africa and by Kanitz et al. (2014) who tracked the Saharan Air layer across the tropical Atlantic with lidar measurements. *In-situ* measurements of dust events have been done along the west African margin (Stuut et al., 2005), Morocco (Kandler et al., 2009) and Cape Verde (Kandler et al., 2011), and with time-series sediment traps on land in Niger, Mali and

Senegal (Marticorena et al., 2010;2011;Skonieczny et al., 2013;Kaly et al., 2015) as well as time-series sediment traps in the ocean off NW Africa (Ratmeyer et al., 1999a;1999b;Friese et al., 2016). Proxy records have been constructed from Cape Verde corals (Mukhopadhyay and Kreycik, 2008), and on much longer time scales in sediments cored from the sea floor (Tjallingii et al., 2008;Mulitza et al., 2010;Filipsson et al., 2011). On the other side of the ocean, Prospero and colleagues have been sampling Saharan dust mainly on Barbados since the late

1960's, resulting in the longest continuous time series of Saharan-dust sampling in the world (Prospero et al., 1970;Prospero and Lamb, 2003;2014). Dust deposition onto South America (Prospero et al., 1981) and its impact on fertilizing the Amazon rain forest was researched for example by Swap et al. (1992);Bristow et al. (2010) and (Yu et al., 2015b)

All observations of Saharan dust showed a strong seasonality with higher dust fluxes during the winter season

close to the dust sources and South America and higher dust fluxes during the summer season in the Caribbean. This is related to the prevailing wind systems, which are influenced by the movement of the intertropical convergence zone (ITCZ). Saharan dust is transported during any time of the year by the northeasterly trade winds at relatively low altitudes, carrying the dust to the proximal parts of the Atlantic Ocean (Pye, 1987;Stuut et al., 2005). During the winter, when the ITCZ reaches its most southerly position, the Harmattan trade winds become

more important. The Harmattan trade winds transport the dust in surface winds at altitudes below three km (Sarnthein et al., 1981) and crosses the Atlantic Ocean in the direction of South America (1981;Prospero et al., 2014). As the ITCZ migrates northward during the summer, the dust is transported at higher altitudes up to five km (Tsamalis et al., 2013), and crosses the Atlantic Ocean above the trade wind zone in the direction of North America and the island of Barbados (Prospero et al., 1970;2014). The sedimentological characteristics of

transported dust reveal coarser-grained dust in summer close to the dust sources (Ratmeyer et al., 1999b;Skonieczny et al., 2013;Friese et al., 2016;Van der Does et al., 2016) and a mineralogical composition of mainly quartz, feldspar, calcite, clay minerals and iron oxides in varying quantities (Caquineau et al., 2002;Kandler et al., 2009).

A 25-year time series of northwest African dust fluxes was established by colleagues at the University of Bremen,

Germany, using sediment traps moored on the Mauritanian continental slope. First results were presented by Ratmeyer et al. (1999a; 1999b) and Nowald et al. (2015), followed by the 25-year record by Fischer et al. (2016). On average, lithogenic particles make up about a third of the total mass flux and up to 50 % during dust events



(Nowald et al., 2015). Biogenic mass fluxes in this area are generally high as well, as the sediment traps are located in one of the four major Eastern Boundary Upwelling Ecosystems (EBUSs) (Fréon et al., 2009), bringing cold, nutrient-rich waters to the surface waters stimulating primary productivity. The biogenic matter consists of organic matter (OM) as well as marine biominerals of $CaCO_3$ and amorphous silica ($BSiO_2$) which are produced by

autotrophic phytoplankton and heterotrophic zooplankton. In particular, phytoplankton $CaCO_3$ producers are mainly coccolithophores while zooplankton $CaCO_3$ is mainly from foraminifera and gastropods shells. The $BSiO_2$ is primarily produced by the phytoplankton diatoms and zooplankton radiolarians. Fischer et al. (2016) demonstrate that the Cape-Blanc sediment-trap series did show a weak relationship between dust input and productivity as reflected by the biogenic silica, mainly derived from diatoms as important primary producers. In

addition, dust particles were found to strongly enhance the settling of organic matter through the water column, although no evidence was found for a relation between bulk fluxes and dust particle size. They further suggest that precipitation may determine dust deposition in summer as opposed to dry deposition of dust in winter. However, particle-size distributions of Saharan dust indicate that other factors such as wind speed and episodic dust events should be taken into account as well (Friese et al., 2016). Throughout the last 25 years, the bulk fluxes off Cape

Blanc varied strongly on seasonal and interannual timescales but without any significant relationship to known large-scale climatic oscillations (Fischer et al., 2016). As their sediment traps are located relatively close to the Mauritanian continental shelf, likely also lateral input from re-suspended shelf sediments was collected, propagating down the slope in nepheloid layers (Fischer et al., 2009).

On the other side of the Ocean, the Barbados time-series revealed an apparent relationship between Saharan dust

transport to the island and droughts in the Sahel during the previous year (Prospero and Nees, 1977). However, the robustness of this relation decreased the longer the record became from $R^2=0.92$ after ten years through $R^2=0.73$ after 2years to $R^2=0.55$ after thirty years (Prospero and Nees, 1986;Prospero and Lamb, 2003). The authors argue that not only droughts in the source regions play a decisive role in the dust emissions from Africa but also meteorological conditions during transport and deposition should be taken into account. The composition of the

dust arriving on Barbados is rather uniform and represents the concentration of the upper continental crust, indicating either similar source areas or different source areas with subsequent mixing during transport (Trapp et al., 2010).

In order to determine the transport and deposition of Saharan dust between the source in Africa and the sinks in the Atlantic- and Caribbean Oceans, we deployed an array of five moorings below the core of the dust plume,

starting in 2012. Based on eight years of satellite observations, the Saharan dust plume is very consistently located between 7° and 17°N (Mulitza et al., 2008). Therefore, the array of moorings was positioned along the 12th northern parallel between 23°W and 57°W, each equipped with two time-series sediment traps. In addition, we positioned a land-based dust collector in Iwik on the Mauritanian coast nearest to the source (Fig. 1). Here, we present the initial set-up of the monitoring experiment and the first-year results of the mass fluxes and their composition as

intercepted by the sediment traps in the Atlantic and the land-based samplers in Iwik. The results are compared both with each other, and satellite images of atmospheric Saharan dust transport. Complementary results of the size distributions of Saharan dust from the sediment traps are discussed by Van der Does et al. (2016).

## 2 Instrumentation and Performance

### 2.1 Land-based dust collectors

Saharan dust transport and deposition is traced from source to sink, starting on the Mauritanian coast with two masts with passive modified Wilson and Cooke samplers (MWAC dust collectors; Goossens and Offer (2000),





located in Iwik, ($19^0$53.11'N, $16^0$17.64'W) which have been sampling dust over monthly intervals since January 2013. Each mast contains a total of ten MWAC samplers, deployed in pairs at five heights (0.9 – 2.9 m) from the ground. The masts contain a wind vane that directs the opening of the MWAC collectors into the wind.

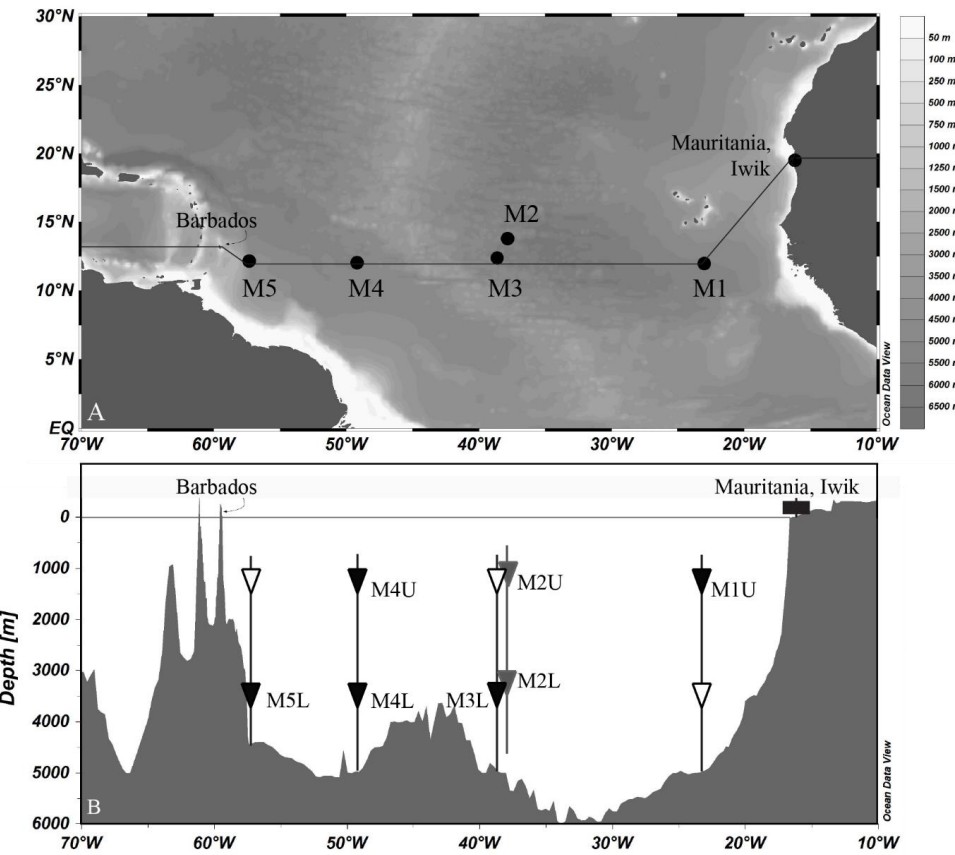

**Figure. 1. A: Location of land-based dust collectors on the Mauritanian coast in the village of Iwik and sediment trap moorings M1 – M5 between northwest Africa and Barbados. B: Bathymetric profile through the equatorial North Atlantic between Mauritania (15° W) and Barbados (60° W) showing the five moorings with sediment traps (triangles) and the land-based MWAC dust collectors (rectangle). MxU and MxL indicate the upper and the lower traps in the five moorings at 1200m and 3500m nominal water depth, respectively. Open symbols refer to sediment traps that failed or were lost during recovery.**

**2.2 Ocean moorings**

Moorings were deployed at five sites (M1 – M5) along a transect in the equatorial North Atlantic during RV Meteor cruise M89 (Stuut et al., 2012) in October 2012 (Fig. 1). Four of the five moorings (M1, M3, M4 and M5) were deployed at $12^0$N and $23^0$W, $38^0$W, $49^0$W and $57^0$W, respectively, and one mooring (M2) was positioned to the north of station M3 and deployed at $13.5^0$N and $37.5$ $^0$W to assess potential north-south movements of the Saharan dust plume. Each mooring was equipped with a number of oceanographic instruments (Table 1). These include two Technicap PPS 5/2 sediment traps provided with a tilt meter at a nominal water depth of 1200 m (upper) and 3500 m (lower), two SBE MicroCat CTDs for conductivity, temperature and depth measurements, two Aanderaa RCM-11 current meters, and four floats to keep the mooring upright. The uppermost float included





a downward-looking Acoustic Doppler Current Profiler (ADCP, 75 Hz) for measuring current profiles (velocity and direction) and particle backscatter intensities, and a XEOS iridium beacon and flasher on top.

**Table. 1. Layout of moorings' instruments as an example of sampling site M1**

| Instrument name | Description | Water depth (m) |
|---|---|---|
| Benthos floats | Floats | 720 |
| Smartie float 500 | Float with beacon + flasher + downward looking ADCP | 750 |
| SBE MicroCat CTD | CTD – Conductivity Temperature & Depth sensor | 1190 |
| Technicap PPS 5/2 | Sediment trap with 24 bottles + tilt meter | 1200 |
| Smartie float | Float with 500kg buoyancy | 1250 |
| Aanderaa RCM | Current meter | 3480 |
| SBE MicroCat CTD | CTD – Conductivity Temperature & Depth sensor | 3490 |
| Technicap PPS 5/2 | Sediment trap with 24 bottles + tilt meter | 3500 |
| Benthos floats | Floats to retrieve mooring | 40m above bottom |
| Aanderaa RCM | Current meter | 20m above bottom |
| Releasers | To detach mooring from anchor | 10m above bottom |
| Anchor | Steel weight 2000kg | bottom |

Generally, current velocities did not exceed 12 cm s$^{-1}$, which is considered a threshold for unbiased collection of settling particles (Knauer and Asper, 1989). Current-meter measurements showed that the average velocities around each mooring were < 6 cm s$^{-1}$ in the deep and bottom ocean at around 3500 and 4600 m water depth, respectively and < 10 cm s$^{-1}$ in the upper ocean as measured by the ADCPs at around 1200 m water depth. For only a few days during the entire sampling period current velocities exceeded the 12 cm s$^{-1}$ for the upper sediment traps at 1200 m at the sites M2 and M3, while at M4 maximum velocities at 3350 m just reached 19 cm s$^{-1}$ in early August 2013. All other sensors showed that the sediment traps at sites M1 to M4 remained well within 5$^0$ from the vertical and at constant depths during the entire sampling period. Only during two periods at station M5 current velocities exceeded 12 cm s$^{-1}$ to the extent of biasing collecting efficiency: in February 2013 and late March 2013. This caused a downward movement of the traps as recorded by increased pressure (depth) by both CTDs, as well as deviations of up to 14$^0$ and 12$^0$ from the vertical, respectively, as measured by the tilt meters, affecting three sample intervals at station M5 (intervals #8, 10 and 11). For these three intervals we adjusted the total mass flux using $F = F_t - (1+1.4\sin2\theta)$ with F being the vertical flux, $F_t$ the flux in the tilted trap, and $\theta$ the degree of tilt from the vertical (Gardner, 1985).

Each sediment trap has a collecting area of 1m$^2$ and is equipped with an automated rotary collector carrying 24 sampling cups. All traps collected the settling particles simultaneously and synchronously in 16-day intervals, starting on October 19, 2012 and ending on November 7, 2013. Prior to deployment in 2012, sampling cups had been filled with filtered seawater from the respective sampling depths at 1200 m and 3500 m, using HgCl$_2$ as a biocide (to an end concentration of 1.3 g/L) and Borax (Na$_2$[B$_4$O$_5$(OH)$_4$] *10H$_2$O; end concentration 1.3 g/L) as a buffer at pH 8.5. Both additives increased the density to slightly higher than seawater to limit exchange with ambient seawater. Seven sediment traps were recovered successfully during RV Pelagia cruise 64PE 378 in November 2013 (Stuut et al. 2013). These include three upper (1200 m) sediment traps at sites M1, M2 and M4 and four lower (3500 m) sediment traps at sites M2, M3, M4 and M5 (Fig. 1, Table. 2). On board, the pH of the supernatant solution was measured and an aliquot of supernatant liquid was analysed for nutrient concentration (SiO$_4^{4-}$, NH$_4^+$, PO$_4^{3-}$) for shipboard quality control. When necessary, samples were post-poisoned and brought to





an appropriate pH when lower than 8 to prevent $CaCO_3$ dissolution. Samples were kept dark and cool at $4^0$C until further processing.

**Table. 2. Sediment trap details. Sampling was performed simultaneously and synchronously from October 19, 2012 until November 7, 2013 (384 days). (mbsl = meter below sea level, MAR = Mid-Atlantic Ridge)**

| Trap | Position | Bottom depth (m) | Trap depth (mbsl) | Distance to African coast (km) | Closest horizontal distance to seafloor at trap depth (km) |
|------|----------|------------------|-------------------|-------------------------------|------------------------------------------------------------|
| M1U | 12˚00'N, 23˚00'W | 5000 | 1150 | 700 | 610 |
| M2U | 13˚81'N, 37˚82'W | 4790 | 1235 | 2300 | 2260 |
| M2L | | | 3490 | | 520 (MAR) |
| M3L | 12˚39'N, 38˚63'W | 4640 | 3540 | 2400 | 500 (MAR) |
| M4U | 12˚06'N, 49˚19'W | 4670 | 1130 | 3500 | 640 |
| M4L | | | 3370 | | 580 (MAR) |
| M5L | 12˚02'N, 57˚04'W | 4400 | 3520 | 4400 | 63 |

## 3 Methods

### 3.1 Particle Mass Fluxes

From the land-based dust collector, all dust was removed from each sample bottles and weighed on a micro-balance for every bottle height. Dust fluxes represent the horizontal transport fluxes of Saharan dust in the source. It is estimated by equation (1)

$$F_{MWAC} = \frac{MAR}{A} * \frac{1}{n} \tag{1}$$

where F is the dust flux (g m$^{-2}$ d$^{-1}$), MAR is the mass accumulation rate (g d$^{-1}$), A is the cross-sectional area of the inlet tube of the MWAC sampler (m$^2$) and n is the estimated sampling efficiency of the MWAC bottles. The sampling efficiency of the MWAC samplers is 90 % (Goossens et al., 2000)

For the marine particle fluxes from the sediment traps, the samples were wet-sieved over a 1 mm mesh, wet-split in five aliquot subsamples using a rotary splitter (WSD-10, McLane Laboratories), washed to remove the $HgCl_2$ and salts, and centrifuged. Afterwards samples were freeze-dried and ground. Total mass fluxes refer to the < 1 mm size fraction and were determined by weighing two freeze dried 1/5 aliquots for every sample. Average weights between replicate aliquots were within 2.4% (SD = 2.2) and less than 12%, with 87% of all samples differing < 5% between splits. Total mass fluxes were determined by equation (2)

$$F = MAR * A^{-1} * d^{-1} \tag{2}$$

where F is the total mass flux (mg m$^{-2}$ d$^{-1}$), MAR is the mass accumulation rate (mg), A is the sediment trap funnel opening (m$^2$), and d the sampling time interval.



Total nitrogen (TN), total carbon (TC) and organic carbon (TOC) content were determined with a Thermo Scientific Flash 2000 Elemental Analyser. Samples for TOC measurements were decalcified by acid fuming with a subsequent addition of 2 N HCl and dried in an oven at $60^0$ C. Samples for TN and TC measurements remained untreated. Carbonates were calculated as $CaCO_3 = (TC - TOC) * 8.33$ and organic matter as $OM = 2*C_{org}$.

Conversion factors for calculating organic matter vary from 2.0 in the eastern Atlantic Ocean to 2.5 in upwelling areas (Jickells et al., 1998;Klaas and Archer, 2002;Thunell et al., 2007;Fischer and Karakas, 2009); we chose to use the factor of 2 for better comparison of particle fluxes influenced by Saharan dust deposition off Cape Blanc (Wefer and Fischer, 1993;Fischer et al., 2007;Fischer and Karakas, 2009)

Biogenic silica (BSi) was analysed by sequential alkaline leaching on a HITACHI U-1100 spectrophotometer after

Koning et al. (2002). Briefly, 25 – 30 mg of ground sample was leached in a 0.5 M NaOH solution and subsequently reacted with a sulphuric acid – molybdate solution. This solution was flushed through a photocell where absorption at 660 nm was recorded every second. Each sample was run for 60 to 90 minutes. Results were evaluated with a weekly measured standard calibration curve ($R^2 > 0.99$) and calculated with the MS Excel data solver tool. The diatom reference material (pure *Thalassiosira punctigera* from the North Sea) is measured with a

reproducibility of ±0.46 % and sample reproducibility is ±0.36%. For estimating mass fluxes, BSi is expressed as $BSiO_2$ (BSi*2.139), although this conversion systematically underestimates the actual mass by 10-20 % given the crystal water associated with the opaline silica of which the siliceous plankton consists (Mortlock and Froelich, 1989).

The remaining, residual mass fraction is often referred to as the lithogenic fraction (Wefer and Fischer,

1993;Fischer and Wefer, 1996;Fischer et al., 2016); since it contains all the refractory lithogenic particles (quartz, clay minerals, feldspars). The residual mass is defined by subtracting the biogenic $CaCO_3$, $BSiO_2$ and OM from the total mass (Eq. 3) and includes all Saharan dust:

$$\text{Residual mass} = \text{Total mass} - CaCO_3 - BSiO_2 - OM \qquad (3)$$

However, this residual fraction potentially also includes biogenic phosphates and sulphates, as well as particles of

volcanogenic, cosmogenic and anthropogenic origin. In addition, crystal water associated with the opaline silica and the O-H content of organic matter and clay minerals also contribute to the residual mass.

### 3.2 XRF element analysis

The elemental composition of each sediment trap sample was determined by X-ray fluorescence (XRF) using the Avaatech XRF core scanner (Richter et al., 2006). This analytical technique has the important advantage that it is

non-destructive, allowing that very small-size samples – such as sediment trap samples – can be used for other analyses after measurement. XRF scanning results in semi-quantitative compositional data, being expressed as intensities (i.e. counts or counts per second), which we normalize to the total counts to take into account the closed sum of geochemical data. Similar to XRF scanning of down-core sediment samples (Bloemsma et al., 2012), we analyse our data qualitatively as normalized element intensities, which relate to the relative concentrations of these

two elements (Weltje and Tjallingii, 2008).

Ground sediment trap samples (~25 mg) were pressed in polyethylene cylinders with a circular recess of 6 mm and 1.5 mm depth and covered with SPEXCerti Ultralene® foil. This way we are largely avoiding common XRF-scan complications relating to measurement geometry (e.g. interstitial water content and large grain sizes) (Weltje and Tjallingii, 2008). All samples were measured with a 4x4 mm slit size at a voltage of 10, 30 and 50 kV

(elements: Al, Si, K, Ca, Ti, Cr, Mn, Fe, Cu, Zn, Sr, Zr and Ba), with an electric current of 1.5, 1.3 and 0.8 mA, respectively, and a measurement time of 20, 40 and 80 seconds, respectively. All measurements were performed





five times, and we show the average values for these measurements. The elements Ti, Al, Fe and K were chosen and shown to represent lithogenic minerals, while Si represents both lithogenic Si minerals and biogenic produced silica. Pearson correlation was applied to the whole dataset of which we show the five elements Ti, Al, Fe, Si and K in table S1. Five of the sediment trap samples (M1U #9, 12 and 24; M4U #12 and 24) could not be analysed due

to insufficient material.

Two certified external standards (powdered marine sediment MESS-3 and PACS2) were used for quality control of the XRF measurements, and were both analysed within each run of ~25 samples in exactly the same way as the sediment trap samples. These standards showed a precision, expressed as relative standard deviation (RSD = standard deviation / average of replicates x 100%), < 7 % for all normalized element intensities.

**4   Results**

Horizontal transport fluxes from the land-based MWAC sampler are given in g m$^{-2}$ d$^{-1}$ (Fig. 3). The vertical (downward) deposition fluxes from seven sediment traps deployed across the Atlantic Ocean (Fig. 1, Table 2) are treated in downwind succession from east to west, starting from ocean site M1 closest to Africa, to ocean site M5 closest to the Caribbean. For each trap, results are given for total mass flux, the biogenic components CaCO$_3$,

BSiO$_2$, OM, and the residual mass fraction, plus the relative contribution for each constituent as daily (annual) averages in mg m$^{-2}$ d$^{-1}$ (g m$^{-2}$ a$^{-1}$) and annual relative fluxes (Fig. 4). The seasonal variation is shown for specific fluxes as deviation from the annual relative mean, together with the grain size distribution of Saharan dust on the same material (Fig. 5). Saharan dust fluxes are characterized by the normalized intensities of lithogenic elements (Ti, Al, Fe and K), that are each highly correlated with the residual mass fraction (Fig. 7, Table S1). The horizontal

mass fluxes of the land-based dust collector in Iwik are almost 100 % pure dust with neglectable contribution of organic matter and diatoms from paleolakes and are used for chemical element comparison to the residual mass fraction from the sediment traps (Fig. 5). Molar C : N ratios of the marine organic material are compared for the upper and lower traps (Fig 6).

**4.2.1 Horizontal land-based transport fluxes**

The total mass fluxes of the MWAC samplers show significant month-to-month differences (Fig. 3). Highest dust fluxes of around 400 and 250 g m$^{-2}$ d$^{-1}$ were found during spring in March and April 2013 and during summer in July and August 2013, respectively.

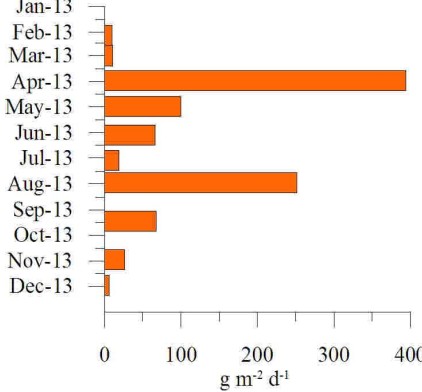

**Figure. 3. Monthly resolved transport fluxes of Saharan dust from the land-based MWAC sampler in Iwik,**
**Mauretania (19°N, 16°W) between January and December 2013.**





### 4.2.1 Mass fluxes at ocean site M1

The shallow sediment trap at 1150 m water depth at site M1 collected a fairly steady total mass flux year round (Fig. 4) with a daily average of approximately 100 mg m$^{-2}$ d$^{-1}$ which equals an annual export of 40.7 g m$^{-2}$ a$^{-1}$. The mass fluxes of the marine biogenic constituents (CaCO$_3$, BSiO$_2$ and OM) follow those of the total mass and are in

close relation to each other. The composition of the trapped material is dominated by CaCO$_3$ (21-45 %) and the residual mass fraction (30-58 %), with minor OM (14 %) and BSiO$_2$ (10 %). A clear seasonal pattern in the total mass flux is absent, whereas a peak flux is recorded in March (interval #9). Relative to the annual mean, biogenic matter fluxes are high during winter and spring, and low during fall and especially summer (Fig. 5). Consequently, the residual mass fraction shows the opposite, with a low relative abundance in winter and spring and a high in

summer and fall. A pronounced seasonality is also seen in the grain-size distributions of the same material with coarser grained Saharan dust in summer and finer grained dust in winter (van der Does et al. 2016). The residual mass flux ranges from 32-71 mg m$^{-2}$ d$^{-1}$ with an average of 47.7 mg m$^{-2}$ d$^{-1}$ (17.4 g m$^{-2}$ a$^{-1}$) and slightly elevated values during summer and fall but with a peak in March.

### 4.2.2 Mass fluxes at ocean site M2

Total mass fluxes at site M2 average 40 mg m$^{-2}$ d$^{-1}$ (14.4 g m$^{-2}$ a$^{-1}$) in the upper trap at 1235 m and 55 mg m$^{-2}$ d$^{-1}$ (19.9 g m$^{-2}$ a$^{-1}$) in the lower trap at 3490 m (Fig. 4). Peak fluxes are recorded during spring and summer, especially in March and late April (intervals #9 and 12). The composition of the material is rather similar in the upper and lower trap. In general, the trapped material is dominated by CaCO$_3$ (56 % and 59 %, respectively), followed by the residual fraction (27 % and 28 %, respectively), OM (11 % and 7 %, respectively) and BSiO$_2$ (both about 6

%). No clear seasonal pattern is seen in either traps in the total mass fluxes despite some peaks which occur in the upper and the lower trap during the same sampling intervals (Fig. 5). No clear seasonal pattern appears in the biogenic fractions, whereby CaCO$_3$ and BSiO$_2$ tend to be relatively enhanced during spring and summer and OM in fall. Therefore, the resulting residual fraction is relatively enriched during summer and fall and suppressed during winter and spring. The small contribution of the residual fraction in spring is also expressed in its mass

flux, yet is more variable in the upper trap than in the lower trap. This also holds for the grain-size distributions, while the dust particles in the upper trap show a distinct seasonality with coarser grains in summer and fall, the grain sizes of the dust particles in the lower trap fluctuate with no clear seasonality (van der Does et al. 2016)

### 4.2.3 Mass fluxes at ocean site M3

At site M3 about 220 km south of site M2, the total mass flux at 3540 m water depth amounts to 70 mg m$^{-2}$ d$^{-1}$

(14.4 g m$^{-2}$ a$^{-1}$). A double peak flux occurs during summer (intervals #15 &16). The peak fluxes are mainly CaCO$_3$, which forms more than half of the total mass (52 %) followed by OM (8 %) and BSiO$_2$ (7 %). Around one third (33 %) of the total mass resides in the residual fraction, with maxima during fall and minima during spring. Seasonality is weak but several small peaks appear, especially in June but also during spring (March and late April) and during winter in January (Fig. 4). The differences from the mean contribution show enhanced CaCO$_3$ in winter

and spring and BSiO$_2$ in summer and fall 2013 (Fig. 5). The OM shows little seasonal variation, albeit slightly suppressed during spring.

### 4.2.4 Mass fluxes at ocean site M4

Average total mass flux amounts to 85 mg m$^{-2}$ d$^{-1}$ (30.9 g m$^{-2}$ a$^{-1}$) in the upper trap and 64 mg m$^{-2}$ d$^{-1}$ (23.2 g m$^{-2}$ a$^{-1}$) in the lower trap. Both traps intercepted two striking peak flux events, the first during spring (late April) and





the second during fall 2013 (late October/early November), when total mass fluxes exceeded 200 mg m$^{-2}$ d$^{-1}$ in the upper and 100 mg m$^{-2}$ d$^{-1}$ in the lower trap (Fig. 4). The peak fluxes differ in composition, while the peak in spring is CaCO$_3$ dominated (60 % in the upper trap and 73 % in the lower trap), the peak in fall is marked by BSiO$_2$ (25 % and 28 % in the upper and the lower trap, respectively). In addition, both peaks show elevated OM fluxes in both traps. Residual fluxes are elevated in the upper trap as well, but in the lower trap only in fall. The seasonal variability of the grain size distributions become smaller but are still visible in the upper trap with coarser grained dust during summer and fall (van der Does et al. 2016). Seasonal variability of BSiO$_2$ is dominated by the exceptionally high peak in fall (Fig. 5).

**4.2.5 Mass fluxes at ocean site M5**

The sediment trap at 3520 m water depth at the western most ocean site M5, intercepted an average total mass flux of 116 mg m$^{-2}$ d$^{-1}$, which equals an annual mass flux of 42.2 g m$^{-2}$ a$^{-1}$ (Fig. 4). Most of the trapped material resides in the residual mass fraction (45 %), followed by CaCO$_3$ (33 %), BSiO$_2$ (17 %) and OM (6 %). Seasonal variability is clear in the total mass flux with maxima during spring and halfway through summer followed by a changeover in August to lower values during the second half of summer and fall 2013 (Fig. 5). Clearest seasonality is shown by BSiO$_2$ with a pronounced maximum in summer to late fall and minima in winter and spring. Seasonality is less clear in CaCO$_3$ and absent in OM, while the residual mass shows maxima in spring and minima in summer to late fall. The grain sizes of the residual fraction become smallest at the western most station and varies around the clay size fraction (van der Does et al. 2016)







**Figure 4. Mass fluxes at ocean site M1U to M5L in mg m$^{-2}$ d$^{-1}$ (bars) for total mass (black), CaCO$_3$ (white), BSiO$_2$ (grey), organic matter (green) and the residual mass fraction (orange); black lines represent their relative contributions (wt%).**



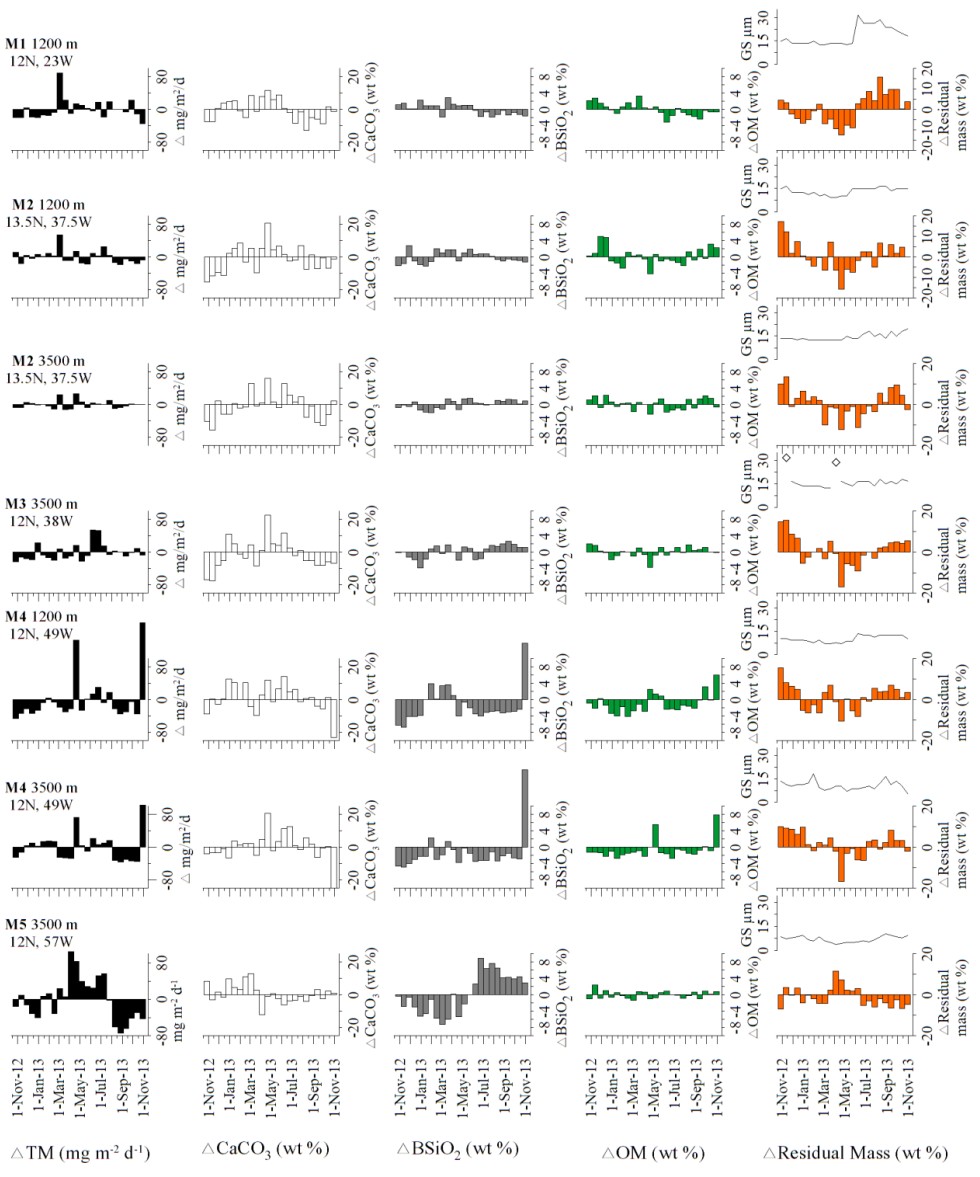

**Figure 5. Deviation from the flux-weighted annual mean for total mass (black), CaCO3 (white), BSiO2 (grey), organic matter (green) and the residual mass fraction (orange) at ocean sites M1U to M5L. Black line represents modal grain sizes of Saharan dust of the same sample (data van der Does et al. 2016)**


### 4.2.6. Molar C : N ratios

Total organic carbon (TOC) and nitrogen (TN) are highly correlated at both trap depths (Fig. 6). The upper traps of the moorings M1, M2 and M4 show higher TOC and TN contents than the lower traps of the moorings M2, M3, M4 and M5. However, two fluxes (interval #13 and 24) at site M4L contain as much TOC and TN as found in the upper traps. The average molar C : N ratio of all traps is on average 9.17 with a standard deviation of 0.95. Overall, the ratios in the lower traps are slightly higher than in the upper traps, but without significant changes. The highest value (10.26) is reached at site M3L and the most uniform ratios between 8.46 and 9.65 are observed at ocean site M1U. The molar C : N ratios are in the typical range of sinking detritus collected in deep sediment traps and are comparable to the material collected off Mauritania but without seasonally differences (Fischer et al. 2016).

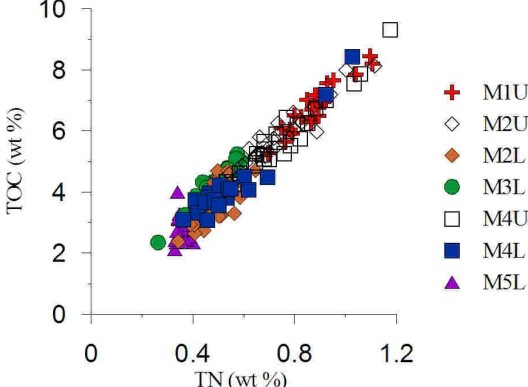

**Fig. 6. Total organic carbon (TOC) versus total nitrogen (TN) content for all ocean sites M1-M5.**

### 4.4 XRF element ratios

Saharan dust is characterized by exclusively lithogenic elements such as titanium (Ti), aluminium (Al), iron (Fe) and potassium (K) that were analysed by powder-XRF. Normalized intensities of these elements are highly correlated with the residual mass fraction and thus are thought to represent the deposition flux of Saharan dust (Fig. 7). However, there are spatial differences from east to west (M1 to M5) relating to the enrichment of especially Al and Fe towards the west.

Normalized intensities of Al and Ti are highly correlated as well, both within the time series in Iwik and throughout all seven sediment traps deployed in the deep ocean (Fig. 8; Table S1, S3). The pure dust sample of Iwik and the samples at the proximal ocean site M1 are most similar in element composition, having the same slope and intercept for Ti and Al normalized intensities (Table S2). Further downwind, Ti/Al intensity ratios are lower but very similar at M2, M3 and M4U, while those at M4L and M5L have very similar but lower slopes (Fig. 8).





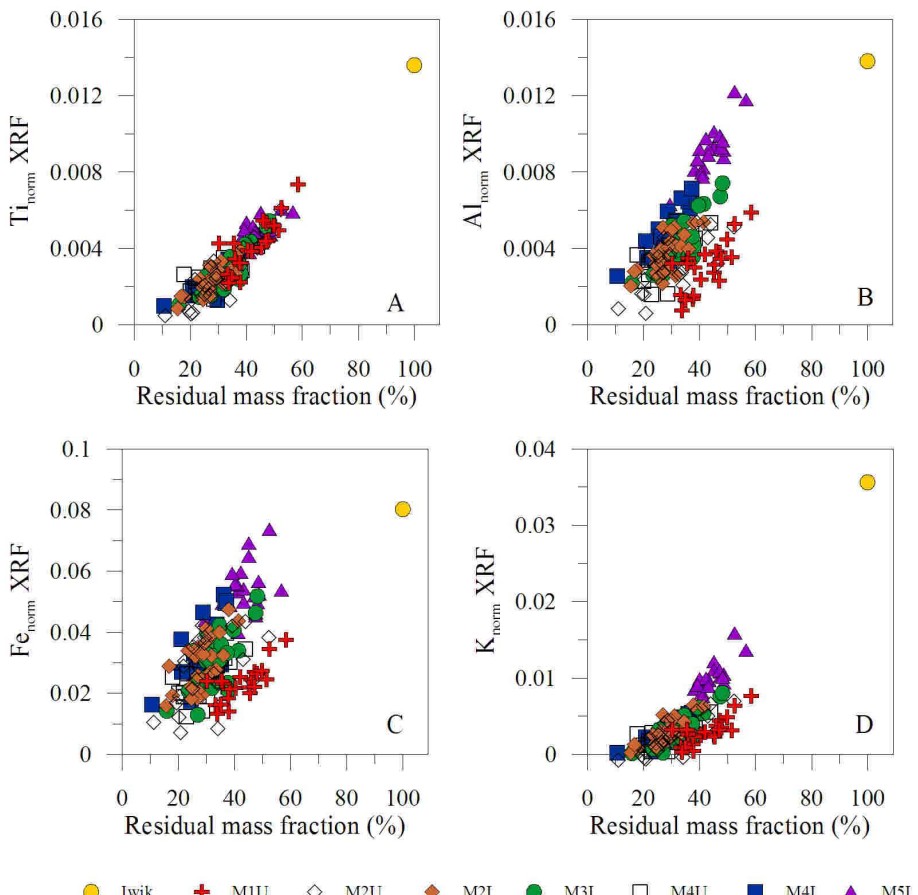

**Fig. 7.** Comparison of the lithogenic elements Ti (A), Al (B), Fe (C) and K (D) normalized to total XRF counts versus the residual mass fraction in all sediment traps and the land-based dust collector in Iwik.

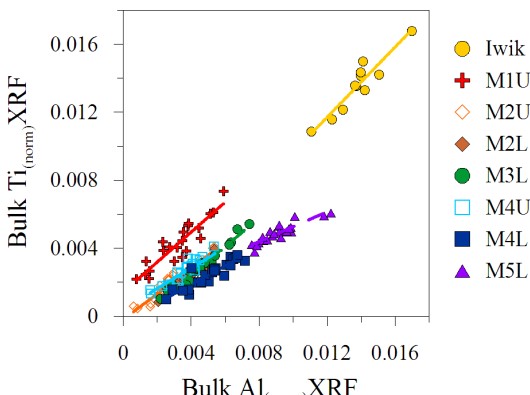

**Fig 8.** Comparison of bulk Ti and Al, normalised to total XRF counts for all seven sediment traps and the
10      on-land dust collectors in Iwik, Mauritania.





Saharan dust mostly consists of lithogenic silicates such as quartz, clay minerals and feldspars (Caquineau et al., 2002) that contribute to the total silica content in addition to the biogenic silica found in the sediment traps derived from skeletal plankton such as diatoms and radiolarian. Regression of biogenic silica determined by sequential leaching to the total silica determined by powder XRF show a clear downwind increase from M1 to M5 (Fig. 9).

Starting at ocean site M1, closest to the Saharan dust source, correlation is weakly negative, then vanishes at M2 and becomes positive and progressively stronger from M3 to M5.

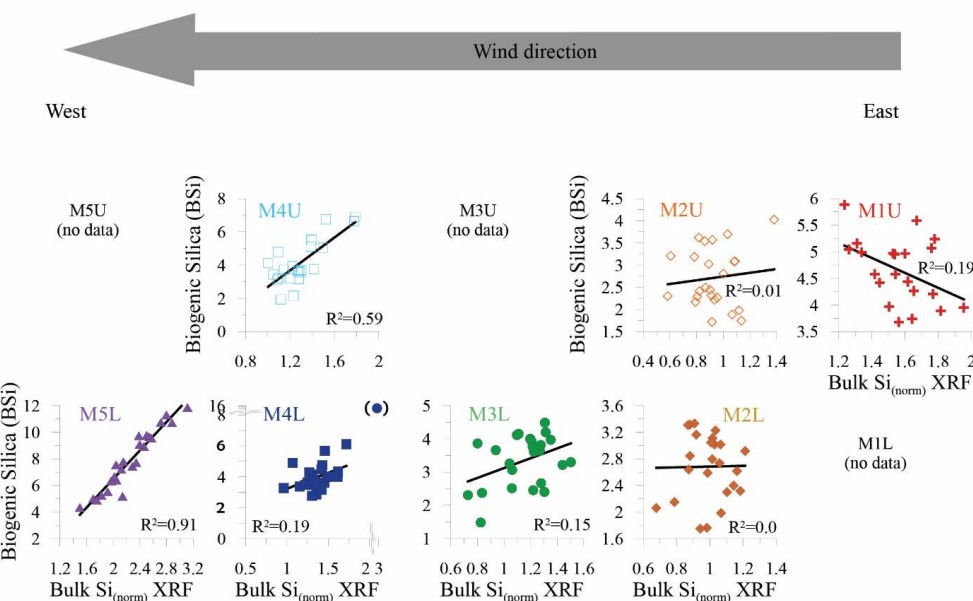

**Fig 9. Biogenic silica versus total silica in particulate fluxes at five sites across the North Atlantic with sampling site M1 in the east (right) and M5 in the west (left).**

**5   Discussion**

Within the DUSTTRAFFIC program we provide the first comprehensive time series of Saharan dust deposition from source to sink across the equatorial North Atlantic Ocean. The land-based dust collector in Iwik is located in the coastal region of western Mauritania (PSA 2), which is one of the major source areas of dust that is transported across the Atlantic Ocean to the Americas (Scheuvens et al., 2013). The transport fluxes are highest in spring and

summer (Fig. 3). The winter high can be related to the trade-wind intensities at lower altitudes, whereas the summer high point to sporadic dust storms, invisible on land by satellites. However, the Saharan dust from the land-based MWAC samplers in Iwik are not only compositionally the same as in the sediment traps at ocean sites M1 in terms of their Ti-Al slopes (Fig. 8) but also testify to a downwind decrease in flux with distance from the source of North African dust. This trend continues downwind as we observe an overall decrease in both the residual mass and the

marine biogenic matter fluxes from ocean sites M1 to M3 to M4. The much higher content of lithogenic elements Ti, Al, Fe and K found in the pure dust at Iwik result from the absence of diluting biogenic matter produced in the marine realm. While the Ti-Al slopes are the same at Iwik and ocean sites M1 through M4U, and Al, Fe and K contents are most similar at Iwik and M1 as well, but become significantly higher towards the west (Fig. 7), it suggests a downwind change in mineralogical composition of Saharan dust. At ocean sites M4L and M5L, both

the Ti/Al slope is lower and the Al content is the same, indicating that a second source in addition to Saharan dust





is involved, which may be derived from admixture of re-suspended clay-sized sediments advected towards the deep sediment traps. This is especially indicated by the offset of K at M5L. Overall, we observe a downwind increase in the Al and Fe content of the residual mass fraction, which suggest an increase in clay minerals relative to quartz. A similar change in mineralogy was observed by Glaccum and Prospero (1980) who compared the mineralogy at the eastern and western Atlantic Islands. Our observation is also supported by the clear downwind trend in the correlation between biogenic silica, as determined by sequential alkaline leaching, and total silica measured by XRF (Fig. 9). Such a downwind decrease in the relative amount of quartz, may result from the slower settling speeds of lighter and platy clay minerals such as micas suspended in the atmosphere. This is consistent with the downwind decrease in grainsize as observed by Van der Does et al (2016) on the same material.

## 5.1 Comparison with satellite observations

Saharan dust needs about a week (5 – 6 days) to cross the Atlantic from east to west as observed in satellite images (Prospero et al. 1970, 2014). Since the sampling interval of all sediment traps is 16 days, it is difficult to identify individual dust outbreaks as individual peaks in mass fluxes that decrease in amplitude from east to west. However, this seems to be the case for the peak fluxes occurring in summer, when there is an enhanced contribution by the residual fraction, especially closest to the African coast at the ocean site M1. Such enhanced fluxes in residual mass are found in intervals #16 and #18 at site M1U and M4U and #16 of M3L, corresponding to the period from 16 June to 2 July, and from 18 July to 3 August 2013, respectively. During these summer intervals, dust outbreaks leave the African coast at high altitudes and propagates westwards across the Atlantic (Fig. 10). Despite this apparently good correspondence between the satellite data and the sediment trap record, there are also other cases where the sediment traps did not seem to record specific dust outbreaks observed in satellite images. This may relate to the altitude at which the dust is travelling; in summer, dust travels through the high atmosphere, as opposed to winter-dust that travels in the low-level Trade zone. The diffuse cloud that is observed across a relatively wide band throughout the atmosphere in summer is interpreted from nadir-looking satellites as higher aerosol optical thickness than the relatively narrow and dense dust cloud that crosses the atmosphere in winter. Conversely, some sediment trap intervals recorded a residual mass peak in spring, e. g. interval #9 which could be followed in M1U, M2U, and M3L, (Fig. 4), when no evidence is found for the occurrence of dust events during that period (24 February – 12 March 2013) in satellite imagery. This could be reflecting the timing of satellite overpasses, which could potentially lead to missing short-term events. However, most dust events that are recorded by satellite images visible as multiple-day events. The presence of clouds in virtually all satellite images is likely to at least partially obscure low-level winter dust outbreaks. Although satellite products are a great help to follow huge dust outbreaks crossing the Atlantic, our results illustrate that there is still a high degree of difficulty in matching specific dust outbreaks observed by satellites and the fluxes we measure in the deep sediment traps. This may also be enhanced by time lag between the deposition of Saharan dust on the ocean surface and their arrival at depth.





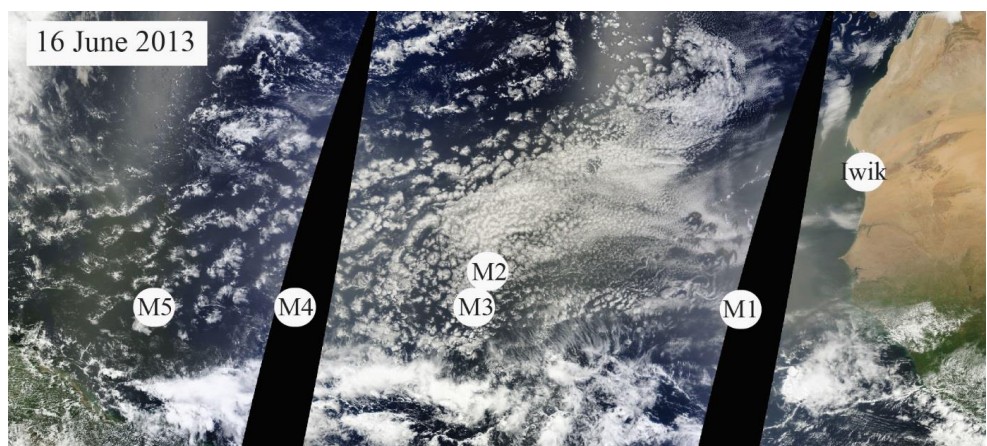

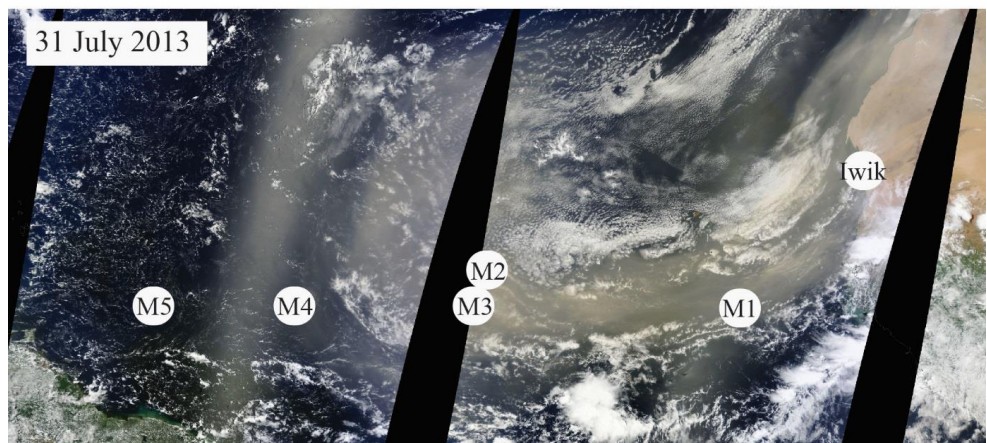

**Fig 10.** Satellite images (EOSDIS Worldview: https:\\worldview.earthdata.nasa.gov) showing dust outbreaks propagating westwards from the African coast across the Atlantic Ocean, the location of the land-based dust samplers at Iwik and the five ocean moorings M1 through M5.

## 5.2 Gravitational settling and downward transport

Saharan dust deposited on the surface ocean is generally too fine to settle out as individual particles, except perhaps for larger and massive quartz particles. Rather, they remain effectively suspended until incorporated by organic matter aggregates such as "marine snow" and fecal pellets that accelerate settling to velocities in the order of 200 m d$^{-1}$ (Knappertsbusch and Brummer, 1995;Berelson, 2002;Fischer and Karakas, 2009) Indeed, at site M4, both the traps at 1200 m and 3500 m intercepted two peak fluxes during the same 16-day interval, one in April and one in October (Fig. 4). Since these traps are 2300 m apart, this results in a settling velocity of at least 140 m d$^{-1}$. Higher settling velocities are conceivable for the fall peak fluxes at station M4, since the TOC and TN content (Fig. 6) of this peak flux in the lower trap is equally high as in all the upper traps (M1, M2 and M4), indicating rapid settling and little degradation. The same is observed at ocean site M2 for the flux peaks in March and in May (Fig. 4). However, the settling pathway might be different, as indicated by the higher total mass fluxes in the lower trap than the upper trap at site M2. Such enrichment might result from the greater catchment area of the deeper trap (Siegel and Deuser, 1997;Waniek et al., 2000). The same holds for the residual mass, which is more enriched in the lower trap compared to the upper trap. Consequently, time lags between Saharan dust outbreaks and transport





in the higher atmosphere as observed in satellite images and their arrival in the upper sediment traps are at least a week and might take another path until arrival in the lower trap two weeks later. In this respect, the buoys deployed now for time-series sampling of atmospheric dust just above the ocean surface at the same sites will serve as an intermediate between satellite observations of dust outbreaks and the actual deposition of dust on the surface ocean,

which can be then followed down by the fluxes in the ocean.

**5.3 Comparison with deep ocean mass fluxes off northwest Africa**

The mass fluxes that we observe in the traps along the DUSTTRAFFIC transect are much in line with observations from sediment traps further north, off Cape Blanc on the Mauritanian continental slope at ~21°N, 20°W. For that site, Fischer et al., (2016) report total mass fluxes of 40.23 g m$^{-2}$ per year averaged over a period of 25 years with

much interannual but little seasonal variability. This compares well with our sampling site M1, where we measured a total mass flux of 40.7 g m$^{-2}$ a$^{-1}$. Cape Blanc residual mass fluxes vary about 5 to 26 g m$^{-2}$ a$^{-1}$ (Fischer et al., 2016), which agrees very well with the average residual flux at site M1 of 17.4 g m$^{-2}$ a$^{-1}$. At both sites the residual mass fluxes make up about a third of the total mass flux and up to 50% during dust events (Nowald et al., 2015). Biogenic mass fluxes in the Cape Blanc area are generally also high, also owing to the upwelling of cold, nutrient-

rich waters that cause high primary productivity in the surface waters. However, biogenic mass fluxes are about the same as found at M1, where no upwelling-stimulated productivity occurs, suggesting Saharan dust may be the common factor fuelling enhanced (export) production.

Mass fluxes collected by sediment traps off Cape Blanc are affected by lateral input from re-suspended sediments that are advected from the Mauritanian continental slope in nepheloid or bottom layers (Fischer et al., 2009).

Similar lateral transport of resuspended sediments (Van Raaphorst et al., 2001;Bonnin et al., 2006) may also have come from the nearby Barbados continental slope and contributed to the high fluxes of the residual mass fraction at ocean site M5 (Fig. 4), which is relatively close to the Barbados margin, around 63 km to the trap depth at 3500m (Fig. 1B, Table 2). This admixture of resuspended sediments is also suggested by the lower Ti/Al ratios and high K contents at M5, and to a far lesser extent also in the deep trap at M4 although residual mass fluxes at

this site are significantly lower (Fig. 4). The enrichment in Al, Fe and K indicates the addition of Al-Fe-K-rich clay minerals from a second source next to the Saharan dust deposited at these sites. This second source of sediments in the deep traps potentially also causes the negative correlation between the residual masses at M5 and M1, while those between M1 to M4 are all positively correlated (Table S3). However, the particle-size distributions at M4 and M5 (Van der Does et al. 2016) do fit the general pattern of decreasing particle size with increasing

distance from the Saharan dust source, thus indicating that the sediments collected by all stations are dominated by Saharan dust.

The particle fluxes at ocean site M5 deviate from the general pattern in terms of enhanced total and residual mass fluxes and seasonal biogenic-silica contribution. Due to its westernmost position, the deep sediment trap may have received considerable amounts of lithogenic or biogenic material originating from either the Amazon- or Orinoco

River, or both. The freshwater Amazon outflow disperses kilometres from the river mouth and affects the oceans' biogeochemistry (Yeung et al., 2012). Due to nutrient input, the occurrence of diatom-diazotroph associations is stimulated (Subramaniam et al., 2008). The fact that the biogenic silica correlates perfectly with the bulk silica (Fig. 9) possibly relates to the appearance of diatom phytoplankton and less contribution of quartz minerals.





## 6 Conclusions

The first-year results of our monitoring experiment yield valuable insights into the transport and deposition fluxes of Saharan dust between the African continent and the Americas. We demonstrate that the lithogenic particles collected in the sediment traps are from the same source as dust collected on the African coast. With increasing distance from the source, lithogenic elements associated with clay minerals become more important relative to quartz settling out closer to the source. The total silica analyses show that the contribution by biogenic silica, produced by marine biota, increases significantly from east to west to the extent that virtually all silica consists of biogenic silica and lithogenic quartz is insignificant in the west furthest away from the African source(s). At the westernmost ocean site, enhanced residual mass fluxes collected by the deep sediment trap suggest admixing with a second source from resuspended ocean floor sediments, although the fining in lithogenic particle size with increasing distance from the source argues against this. Tracing back individual dust outbreaks from satellite images to the arrival in the deep ocean sediment traps is still demanding given sampling resolution and the time lags involved. Best accordance was found in summer, when higher residual mass fluxes are found that are highly co-variant with typically lithogenic elements such as Al and Ti from Saharan mineral dust. While the temporal and spatial variability in residual mass fluxes does correspond with the changing fluxes of Saharan mineral dust, they seem to overestimate the netto fluxes due to underestimation of marine biogenic matter. To better approximate the deposition of Saharan dust, the buoys that were deployed in 2013 should give information on precise fluxes, which can then be followed in the underlying sediment traps.

### Acknowledgements

The Project is funded by ERC (project no. 311152) and NWO (project no. 822.01.008). We thank the crew of Meteor Cruise M89, Pelagia Cruise 64PE378 and NIOZ technicians for their contributions. Camara from the Parc Natianal de Banc d'Arguin (PNBA) is thanked for assistance with the dust sampling in Iwik, Mauritania. Jort Ossebaar is thanked for helping with the EA analysis and Sharyn Ossebaar for assisting with the biogenic silica measurements. XRF-analysis was supported through the SCAN2 program (NWO project no. 834.11.003) and assistance by Rineke Gieles. Furu Mienis is thank for helping with interpreting the current-meter data and assistance in the sediment-trap lab. The authors also acknowledge the MODIS mission scientists and associated NASA personnel for the production of the data used in this research effort.




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
