# Peer review of "Downward particle fluxes of biogenic matter and Saharan dust across the equatorial North Atlantic"

_Atmospheric Chemistry and Physics, 2016_

## Referee Comment (RC1) · Anonymous Referee #1 · 19 Dec 2016

Review of "Compositional changes of present-day transatlantic Saharan dust deposition" by Korte et al.

The present manuscript deals with the changes in Aeolian dust and marine fluxes on transect from Western Africa to the Caribbean. Sediment traps were deployed in several depths in the ocean, as well as a horizontal flux sampler was on the continent. Seasonal cycles are discussed. A lot of data material is presented and set into context of previous literature. The paper was a pleasure to read for clarity. Some spelling errors remain, which should be taken care of by the technical editing. I wonder though, whether the title is appropriate, as much of the manuscript deals rather with total mass fluxes and changes in total mass flux composition (including biogenic, carbonate), but a minor part only is dedicated to Saharan dust composition.

[Figure]

Being on the atmospheric side, I can't comment on the oceanic sediment-specific techniques. A few minor remarks are made below; in particular the XRF data handling should be re-considered.

P6L9: How was the dust removed from the MWAC bottles – liquid suspension or dry?

P9L17: Goossens and Offer (2000) give the efficiency for a size distribution with a mass median of 30 $\mu$m (geometric) (their Fig. 8). Van der Does et al. (2016) show mass median grain sizes between 4 and < 20 $\mu$m for the sediment traps. How was the size distribution in the MWAC samplers? If the mass median was considerably smaller than 30 $\mu$m, a significant overestimation of collection efficiency / underestimation of mass flux would need to be regarded; see Mendez et al. 2016, http://dx.doi.org/10.1016/j.aeolia.2016.02.003

P7L33-35: As the intensity readings are used and it is calculated with, I would suggest not calling this qualitatively. It does not become clear from the manuscript, whether the procedure of Weltje and Tjallingii (2008) was performed (i.e. log ratios were calculated, relative detection efficiency and specific matrix calibration for the particular setup were obtained), or whether the raw intensity readings were used.

P7L34-35: "these two elements": Weltje and Tjallingii suggest a method based on two elements. If you normalize to the total counts here, you do right that, what Weltje and Tjallingii try to avoid by using two different elements. Please consider revising the data processing or write more clearly, if the method of Weltje and Tjallingii is used.

P8L25: If there were wind speed measurements available at the MWAC station, it should be considered giving the information together with the horizontal mass flux, as averages or as times, where the wind speed was higher than an appropriate deflation threshold. That way, it could be estimated, whether emission was locally dominated.

P9L3: This is actually an import.

P9L15 and P9L38: If there are significant differences between the upper and lower

traps recorded: where does the additional mass come from, where does the lost mass go to? The differences are not small.

Fig. 4 is on the small side. In particular the relative mass percent are difficult to compare. I suggest having an additional column of cumulative/stacked plots in white/gray/green/orange instead of the small separate plots. I would also suggest dividing the x axes by seasons instead of 2-month intervals, as seasons are discussed in the text. Moreover, vertical grid lines at the season boundaries would surely help to read particularly the upper plots.

Fig. 5: What are the diamonds in the modal grain size?

P13L14: "Saharan dust is characterized by exclusively lithogenic elements": I suggest "We identify Saharan dust by elements", as in particular K in the atmosphere might have different sources (biomass burning), which might be not relevant in your case.

P13L19: If the procedure of Weltje and Tjallingii was implemented, the following comment can be neglected, otherwise: Correlating relative readings ('normalized intensities') can produce quickly spurious correlations, in particular, if one of the major elements shows independent variations (e.g., Ca, Si). I highly suggest not doing that, but investigating elemental ratios instead; e.g., Ti/Al vs. Ca/Si.

Fig. 9: What are the units on the y-axis?

P15L13: What does PSA mean?

P15L18: How can the horizontal flux from the MWAC sampler can be compared to the vertical flux of the sediment traps? At least, information on wind direction is necessary here. Also, it seems to me rather improbable that horizontal surface flux on the Northafrican continent should be closely linked to ocean deposition near the Caribbean, as commonly a lot of dust is transported aloft in the Saharan air lay for longer distances (as detailed below in the manuscript).

P15L20,25: By the notion 'content' in relation with XRF, reference is made to a linear

contribution (mass, volume, moles). This seems in contradiction to the method section, where normalized intensities are referred to, which do not reflect directly a mass without proper calibration.

P16L7: Particle size would play the primary role (over shape) in settling speed, so it should be termed like that instead of the ambiguous "lighter". Moreover, the silicium-rich quartz and feldspar particles are usually found at larger particle sizes, which can be extracted from the referenced literature, e.g. the Kandler et al. (2009) paper. As result, this compositional change is consistent with the downwind fining.

---

## Referee Comment (RC2) · Anonymous Referee #2 · 6 Jan 2017

Korte et al. present an interesting account of the total mass and compositional fluxes of African dust across the Atlantic Ocean. This study is complimentary to the size-dependent transport study by van der Does et al. (2016). Generally, the authors report higher concentrations of mass closest to Africa and the Caribbean, yet the composition changes as dust is transported downwind, indicating different dust sources of the study area. Although the authors present a unique story for ACP, there are several issues that need to be addressed prior to publication in its final form.

General comments:

Previous studies have demonstrated atmospheric transport of North African dust is strongest in the summer (Jun – Aug), which is not consistent with the current work. Can the authors comment on why this inconsistency exists?

Chiapello, I., and C. Moulin (2002) TOMS and METEOSAT satellite records of the variability of Saharan dust transport over the Atlantic during the last two decades (1979–1997), Geophys. Res. Lett., 29(8), doi:10.1029/2001GL013767.

Perry, K. D., T. A. Cahill, R. A. Eldred, D. D. Dutcher, and T. E. Gill (1997) Long-range transport of North African dust to the eastern United States, J. Geophys. Res., 102(D10), 11225–11238, doi:10.1029/97JD00260.

The authors should more clearly link the spatiotemporal heterogeneity in the composition of the current work to the size-dependent depositional trends of van der Does et al. (2016), i.e., the smaller clays are transported farther while the larger quartz are deposited closer to Africa.

The first paragraph of the introduction contains a nice summary of previous work, but it reads somewhat like a list of dust observation references with reporting their main findings. The first few sentences do contain this information, stating the quantities of dust measured, but what about the latter half of this paragraph? In order to demonstrate how this study is an advancement of and unique from previous work, discussing what these previous studies found is useful. Additionally, the introduction, in general, is lacking broader implications for evaluating Saharan dust transport. Without the broader impacts, the motivation behind this work is not evident.

Clarification on some of the methods and calculations is needed, specifically pertaining to biogenic Si, the experimental setup with the mast and moored trap locations, and horizontal flux transport. See specific comments below:

Was $BSiO_2$ measured in the Iwik samples to serve as a control; to truly indicate that this Si was of a marine biogenic origin? How was this silica discernable from the mineral dust Si? What would be helpful is if the methods for extracting and detecting BSi contained more explanation, especially since this is not a technique one commonly encounters in an atmospheric journal. For instance, why the 660 nm laser; does it differentiate biogenic from mineral Si somehow, or is there something done in tandem

to the diatom reference material? How does this differ from, say, XRF, in terms of measuring Si? More explanation would be helpful for someone who is not familiar with this technique.

Horizontal flux transport indicates transport over a set distance, however, the authors present this factor as g m2 d-1, which indicates a time-dependent flux. I would assume both masts would be used to calculate the true horizontal flux of dust. Perhaps the wording is throwing me off here, but it seems as if this calculation is at one MWAC sampler as indicated on p8, l11, yet there were 20 samplers (2 masts with 5 pairs of samplers each). How far apart were these masts? Additionally, can the authors comment on how much of the dust flux is missed at higher altitudes, since the samplers extend up to 2.9 m but dust transport can happen up to 5 km?

For those of us not familiar with sediment traps, why were they located so deep and not closer to the surface where the dust would initially deposit and where the Northern Equatorial Current would carry dust from east to west? It seems like there would be a lot of room (literally) for different processes to remove the dust during sedimentation, like the deeper circulation from the conveyor belt in that region (N-to-S directionality). Since many readers will not have an oceanographic background, but are well versed in atmospheric transport mechanisms and how sensitive aerosol transport can be due to shifts in large and small scale circulation patterns, it would be helpful to provide a few sentences early on regarding Atlantic Ocean circulation and how this could potentially impact samples collected 2-3 km deep where the traps were located.

For the calculations of carbonates and OM: Where does the 8.33 correction factor come from for CaCO3? For OM, it seems like this calculation is likely underrepresenting total OM. Why isn't the TON included in OM? There is more than just C in organic matter (O, S, H, N to name a few; see references below as a couple examples), so if these aren't accounted for, the authors should clearly state the limitations and caveats with this calculation. And lastly, what is Corg and how was this measured/calculated (is it just TOC)?
Hazem S El-Zanan, Barbara Zielinska, Lynn R Mazzoleni & D. Alan Hansen (2009) Analytical Determination of the Aerosol Organic Mass-to-Organic Carbon Ratio, Journal of the Air & Waste Management Association, 59:1, 58-69, DOI: 10.3155/1047-3289.59.1.58.

L. M. Russell (2003) Aerosol Organic-Mass-to-Organic-Carbon Ratio Measurements. Environmental Science & Technology 37, 2982-2987.

Some clarification on the elemental components used for mineral classification is needed. For determining the elemental compositions of lithogenic and biogenic minerals, why is Ca not used? Wouldn't this corroborate the carbonates calculation from TOC and TC? For K, was biomass burning K removed? Central Africa is prone to fires and thus a large source of biomass burning aerosol (K is commonly used as a biomass burning marker). This could simply be resolved by calculating soil K, where [non-soil K] = [K] – 0.6[Fe] and [soil K] = [total K] – [non-soil K] from Kreidenweis et al. (2001). Kreidenweis, S. M., Remer, L. A., Bruintjes, R., and Dubovik, O. (2001) Smoke aerosol from biomass burning in Mexico: Hygroscopic smoke optical model, J Geophys Res-Atmos, 106, 4831-4844.

Along these lines, what is the possible contribution from sea salt minerals (i.e., K) recrystallizing during sample processing to the elemental concentrations? Even though the samples were washed for salts, could the authors comment on how salts could potentially recrystallize onto insoluble particles during processing?

None of the figures show uncertainties/error bars. For instance, the methods section says the total mass fluxes (MWAC) have a 90% collection efficiency, so that remaining mass not collected should be accounted for in the measurement uncertainty. In order to clearly draw statistically significant relationships between the months, error bars should be provided.

The seasonal maxima of the MWAC versus moored traps are offset. Can the authors comment on this in the discussion section?
Specific comments:

P1, l17: The way this is written, it sounds like the authors are stating that the African coast is the source. Perhaps change to "African continent" or "North Africa".

P6, l8: To be clear up front, it would be helpful to provide a statement saying which analyses were conducted on the MWAC and which were conducted on the moored trap samples.

P8, l2: State that lithogenic minerals are clays, quartz, and feldspars here. These are in fact listed but much later on.

P8, l6: Please provide the chemical compositions/components of these standards.

P8, l11-23: The addition of all the figures in parentheses in this paragraph place the figures out of order and cause confusion as they are not fully described (my eye was jumping back and forth between figures and text without enough clarification). It would be easier if references to these figures were eliminated here and remain discussed in the following sections. Additionally, this is a nice summary of what is to come, but perhaps providing a sentence at the end describing how all of these different observations will be used in tandem to develop the main picture "'X" would clearly tie it all together.

P8, l26-27: Only Apr and Aug have the spikes (remove Mar and Jul).

Fig 3.: Swap the y and x axes to align with how time series are shown in the rest of the figures.

P9, l16: Why isn't the lower trap shown? At the very least, provide it in the supporting information.

P13, l14-15: It can also contain Ca in the form of calcite (see Glaccum and Prospero (1980)), although at < 10%. A reference for this statement is needed.

P15, l11: First mention of DUSTTRAFFIC; should be introduced in the methods section.

---

## Referee Comment (RC3) · Anonymous Referee #3 · 12 Jan 2017

Thank you for your invitation to review the manuscript submitted to the ACP journal by Laura Korte et al. This paper reports on an ambitious sediment trap experiment carried out in the Tropical North Atlantic Ocean. This experiment, which was set up in the frame of the "DUST TRAFFIC" (ERC) project, aims at documenting Saharan dust inputs to the Atlantic Ocean from the West African coast all the way to the Caribbean, along the main path for Saharan dust atmospheric transport. Korte's paper presents an impressive amount of data obtained during the first year of this experiment, including particle flux and composition temporal variability obtained from 7 sediment traps deployed at about 1100-1200m or 3400-3500m depths, at 5 mooring locations. The manuscript also includes data obtained on land in a dust source region in Mauritania. This study nicely complements earlier investigations carried out off West Africa and in the Sargasso Sea and provides, for the first time, simultaneous flux and compositional

data across the entire Atlantic ocean. This is, I find, the main originality and interest of this report. In any case, this study represents a remarkable logistical and analytical effort, which is to be commended. One of the interesting outcome of this study is that there appears to be no marked seasonal variability in the downward particle fluxes across the entire Tropical Atlantic (a moderate seasonal variability is apparent on the edges of the transect though, near western and eastern margins, especially in the terrigneous fraction and the biogenic silica, respectively). This study also demonstrates the prominent influence of Saharan dust inputs in the "residual" (i.e., non-biogenic) material along most of the E-W transect (based on a few major element ratios (Al, Fe, K) that compare well with ratios measured in the dust they collected in Mauritania). "Dust" major element composition, still, displays a marked evolution across the ocean, enabling Laura Korte et al. to track the progressive mineralogical (and grain-size) sorting during transport (with the gradual reduction of coarse quartz particles contribution from East to West, as indicated for instance by the non-biogenic Si content). Korte et al. study finally reveals the influence of an additional terrigenous source in the Western Atlantic: major compositional differences are recorded in particles collected near the Caribbean -higher K contents for instance- suggesting a possible influence of fluvial discharges (Amazon and/or Orinoco) in the area. Overall, I find that the paper presents a large number of rare, high quality sediment trap data that will be of great interest to those interested in the Saharan dust transport (including modelers) and its impact on the ocean (and Amazonian) biogeochemistry. I therefore recommend publication after the authors have addressed the minor comments listed below.

Detailed comments

page 2, line 24-25: Very few studies actually measured "fluxes" (in most cases, dust was collected by air filtration), so you may want to rephrase.

page 2, line 31: Sarthein et al. [1981] (which discusses paleo-dust records) might not be the best reference here – Chiapello et al., [1995] for example, which documents present-day seasonal Saharan dust transport in the trade wind levels, would be more

suitable.

pages 2-3, bridging paragraph: Although there is no doubt that the Bremen group has built the longest sediment trap record in this region, providing one of the most valuable trap data time series ever collected –and should be praised for this!– other groups, in the UK and France for instance, have also contributed to ample sediment flux studies in the area (both off West Africa and in the Sargasso sea) within the 90's BOFS and JGOFS programs among other frameworks. Some results obtained in these experiments are actually fully relevant to Korte's study (see below). So, I think it would be useful to provide a more exhaustive list of earlier experiments (at least provide the references).

page 3, line 12-14: I find this sentence somewhat unclear (the "other factors" have implication on grain size? on the dust deposition timing?) – please clarify.

page 3, line 24: "Composition" is a little vague (you may want to specify here what has been characterized in your study: biogenic components ($CaCO_3$, $BSiO_2$, OM...), major elements in the residual (i.e., lithogenic) fraction (Si, Al, Fe...) etc.).

page 6, lines 20-24: You may want to indicate that sediment traps are supposedly collecting the downward (i.e., "vertical") flux, especially since you just mentioned (line 10) that your land-based collector allows to estimate the "horizontal" dust flux.

page 7, line 34: Which "two" elements? Any? (in this case, you may want to remove "these" or rephrase).

page 8, line15: I find "daily averages" somewhat misleading (as it suggest that flux was sample with a daily or even sub-daily resolution); "daily mean" (and annual mean) might be less ambiguous; you may also simplify (e.g., "...each constituent, expressed both in mg.m-2.d-1 and g.m-2.a-1").

4.2.1 to 4.2.5 Mean annual fluxes ("annual averages") for each component of the settling particulate matter ($CaCO_3$, $BSiO_2$, OM, and for the residual fraction) are of great

interest too, and should also be provided for each trap (either in the text or in a table). I find such data are clearly missing in the manuscript. These would also allow discussing how they compare with results obtained at nearby locations in earlier studies (e.g., Bory and Newton, 2000; Bory et al., 2001), which may provide useful clues on the spatial (and/or interannual) variability in the Saharan dust deposition in the tropical Atlantic off West Africa.

4.4 Shouldn't the M5L (and M4L) compositional difference (K for instance) mentioned in this section?

section 5, page 5, lines 14-16: It seems as if you assume that dust collected at Iwik derives exclusively from local sources (during emission processes); couldn't distal sources also contribute (during deposition processes)? This should be discussed I think, especially as Iwik might be on the path of large outbreaks involving numerous source regions in West Africa.

Page, line 19: The fact there is a downwind decrease in the residual mass flux (and marine biogenic matter fluxes) is interesting but these data are not given in the text or in the table (unless I missed them), as I already pointed out above (4.2.1 to 4.2.5); so please provide numbers (for residual mass fluxes especially, but for the other components as well), especially as such decreases are clearly not obvious from the plots! (cf. fig. 4)

page 16, line 3: suggests

page 16, 5.1, line 17: Is it realistic that the event shown on fig. 10 (i.e., from 31 July) could be sampled in the 18 July- 3 August cups, considering the fact that it takes at least a week for the dust to reach the upper traps (as indicated page 18, lines 1-2)? Especially as dust travels at higher atmospheric levels at that time of the year, as you rightly point it out, and so that it may take longer for the dust to settle through the atmospheric column in the first place.

Section 5.1: Overall (for the reason just stated) I find this section should come after section 5.2, since indications on the settling speed in the water column is obviously essential to make the connection between satellite data and water column measurements (as this is actually acknowledged lines 32-33).

page 16, 5.1, lines 22-24: The point you want to make in this sentence (starting by "The diffuse...") is unclear to me. Please clarify and a reference should be provided.

page 17, section 5.2, lines 7-8: This is an important point: as you rightly indicate, a significant part of the dust deposited in the ocean needs indeed to be incorporated with biogenic matter (OM in particular) in order to settle in the water column; for this reason, it would have been interesting to plot the residual flux vs the OM flux for all traps as this may provide useful clues on the interplay between the two in the particle settling processes in the different sectors of the Atlantic ocean (see for instance Deuser et al., 1983; Jickells et al., 1990; Bory et Newton, 2000... etc.).

page 18, section 5.3, first paragraph: As above, the discussion would benefit from the comparison with all existing data (comments on 4.2.1 to 4.2.5).

page 18, section 5.3, lines 15-17: Additional data on ocean productivity (and its seasonal patterns) in the two regions should be provided in order to support such a statement; I would develop the discussion in order to strengthen this hypothesis, or remove this statement, which is too speculative as such.

page 18, section 5.3, lines 28-31: I am not sure I follow the reasoning here: the contribution of an additional "fluvial" (and therefore fine) component on the western side of the transect seems perfectly consistent to me with westward decreasing trend in the grain size. A major fluvial contribution would also be fully consistent with the negative correlation between M1 and M5. So the statement that your grain size data implies that the residual fraction is dominated at all stations by Saharan dust seems to be at odd with the above (and with the last paragraph of page 18 actually).

page 19, conclusions, line 4: "same" source may be little too affirmative and restrictive (replace with ". . .are similar to the dust collected on the African coast"?).

page 19, conclusions, line 10-11: As above, I see no contradiction here.

page 19, conclusions, lines 13-14: This is surprising, and therefore interesting, but the apparent correspondence should be considered with caution (see my comments on page 16, 5.1, line 17).

page 19, conclusions, lines 15-17: It is unclear what [changing] "fluxes" [of Saharan mineral dust] you are referring to (and how it would be calculated). Also, is this under-estimation discussed in the text?

Finally, I think the paper does not really document dust "deposition" as indicated in the title (this will surely be best documented by the buoys deployed within the DUST-TRAFFIC project). So you may want to reformulate (I would suggest something in the line of "Compositional changes of Saharan across the Atlantic as recorded in the water column downward particle flux").

---

## Author Comment (AC1) · 4 Mar 2017

Dear Dr. Schwarz,

Thank you very much for handling the review process of our manuscript. We received three positive reviews from anonymous reviewers with constructive comments improving the readability and understanding of our manuscript. For this, we would like to thank the three reviewers, and we now acknowledge this in the manuscript.

Firstly, we received comments on the title of the manuscript, which is crucial for the public reading. We agree to the reviewers' comments and decided to change the title from: "Compositional changes in present-day transatlantic Saharan dust deposition", to: "Downward particle fluxes of biogenic matter and Saharan dust across the equatorial North Atlantic". Secondly, the introduction of the manuscript was modified, concentrating on the uniqueness of our experimental set-up of simultaneous particle flux sampling across the entire Atlantic Ocean, including Saharan dust particles.

We received comments on the horizontal dust fluxes collected on the Mauritanian coast. Initially, we used these pure dust samples mostly showing that their element composition is very similar to that of the residual (lithogenic) mass fraction in our samples from the deep ocean, i.e. are derived from the same dust source. Relationships between particle size and the meteorology of dust deposition have recently been accepted as an ACP-Discussion paper by Friese et al., (doi:10.5194/acp-2017-131). However, in order for our manuscript to be clear as such, we added meteorological data to the dust fluxes to set these into context.

Furthermore, questions were raised on the XRF method we used to evaluate the element composition of the samples. We agree that we should elaborate on these XRF-analyses, since they provide a well-accepted and much applied easy and quick way to analyse wet sediment cores but have not been applied to dry sediment trap material before.

Additional concerns about the methods, analyses and calculations of the marine particle fluxes were taken into account and the appropriate parts of the manuscript were rephrased accordingly. In addition, more information is now provided to clarify the methods as such, also for a wider scientific audience. Lastly, minor comments were implemented to promote the readability.

Modifications in response to referee #1 are colored in red, to referee #2 in green and to referee #3 in blue. The comments of the reviewers are in normal text and our reply in italics. We hope you agree with us that the manuscript has much improved and merits publication in ACP.

On behalf of all co-authors, yours sincerely, Laura Korte

Please also note the supplement to this comment:

http://www.atmos-chem-phys-discuss.net/acp-2016-1068/acp-2016-1068-AC1-supplement.pdf

**Supplement:**

Dear Dr. Schwarz,

Thank you very much for handling the review process of our manuscript. We received three positive reviews from anonymous reviewers with constructive comments improving the readability and understanding of our manuscript. For this, we would like to thank the three reviewers, and we now acknowledge this in the manuscript.

Firstly, we received comments on the title of the manuscript, which is crucial for the public reading. We agree to the reviewers' comments and decided to change the title from: "Compositional changes in present-day transatlantic Saharan dust deposition", to: "Downward particle fluxes of biogenic matter and Saharan dust across the equatorial North Atlantic".
Secondly, the introduction of the manuscript was modified, concentrating on the uniqueness of our experimental set-up of simultaneous particle flux sampling across the entire Atlantic Ocean, including Saharan dust particles.

We received comments on the horizontal dust fluxes collected on the Mauritanian coast. Initially, we used these pure dust samples mostly showing that their element composition is very similar to that of the residual (lithogenic) mass fraction in our samples from the deep ocean, i.e. are derived from the same dust source. Relationships between particle size and the meteorology of dust deposition have recently been accepted as an ACP-Discussion paper by Friese et al., (doi:10.5194/acp-2017-131). However, in order for our manuscript to be clear as such, we added meteorological data to the dust fluxes to set these into context.

Furthermore, questions were raised on the XRF method we used to evaluate the element composition of the samples. We agree that we should elaborate on these XRF-analyses, since they provide a well-accepted and much applied easy and quick way to analyse wet sediment cores but have not been applied to dry sediment trap material before.

Additional concerns about the methods, analyses and calculations of the marine particle fluxes were taken into account and the appropriate parts of the manuscript were rephrased accordingly. In addition, more information is now provided to clarify the methods as such, also for a wider scientific audience. Lastly, minor comments were implemented to promote the readability.

Modifications in response to referee #1 are colored in red, to referee #2 in green and to referee #3 in blue. The comments of the reviewers are in normal text and our reply in italics. We hope you agree with us that the manuscript has much improved and merits publication in ACP.

On behalf of all co-authors, yours sincerely,

Laura Korte

**Anonymous Referee #1**

**Comment Referee #1:**

The present manuscript deals with the changes in Aeolian dust and marine fluxes on transect from Western Africa to the Caribbean. Sediment traps were deployed in several depths in the ocean, as well as a horizontal flux sampler was on the continent. Seasonal cycles are discussed. A lot of data material is presented and set into context of previous literature. The paper was a pleasure to read for clarity. Some spelling errors remain, which should be taken care of by the technical editing. I wonder though, whether the title is appropriate, as much of the manuscript deals rather with total mass fluxes and changes in total mass flux composition (including biogenic, carbonate), but a minor part only is dedicated to Saharan dust composition. Being on the atmospheric side, I can't comment on the oceanic sediment-specific techniques. A few minor remarks are made below; in particular the XRF data handling should be re-considered.

**Reply to comment:**

*We agree that the title can be clearer, therefore we changed it to:*

**Changes made in manuscript:**

"Downward particle fluxes of biogenic matter and Saharan dust across the equatorial North Atlantic".

**Comment Referee #1:**

P6L9: How was the dust removed from the MWAC bottles – liquid suspension or dry?

**Reply to comment:**

*When we receive the MWAC bottles from Iwik, they are most often also dusty on the outside, and this dust is first removed by rinsing and wiping the bottle from the outside. Afterwards the dust inside the bottles is shaken out by tapping on the bottle avoiding any liquids leading to possible chemical leaching of the dust.*

**Changes made in manuscript:**

We adapted the following sentence in the manuscript:

Page 6, line 17 & 18

"From the land-based dust collectors, all dust was removed from each sample bottle by loosening and shaking the dry dust out of the bottles. The removed dust was weighed on a micro-balance."

**Comment Referee #1:**

P9L17: Goossens and Offer (2000) give the efficiency for a size distribution with a mass median of 30 µm (geometric) (their Fig. 8). Van der Does et al. (2016) show mass median grain sizes between 4 and < 20 µm for the sediment traps. How was the size distribution in the MWAC samplers? If the mass median was considerably smaller than 30 µm, a significant overestimation of collection efficiency / underestimation of mass flux would need to be regarded; see Mendez et al. 2016, http://dx.doi.org/10.1016/j.aeolia.2016.02.003

**Reply to comment:**

*The reviewer is right in this regard. The efficiency of the MWAC samplers dependents on the particle size of collected material and as Mendez et al. (2016) showed, the efficiency decreases with decreasing particle size (PM10 to PM1). The median grain sizes of the dust presented in our manuscript, however, range between 25 and 50 µm with an average of 38 µm. Therefore, we rely on the efficiency range of 75 to 90 % given by Goossens and Offer (2000). More details on the grain-size distributions of the MWAC samples are now available in Friese et al., 2017 (ACPD).*

**Changes made in manuscript:**

We adapted the following sentence in the manuscript:

Page 6, line 24 – 26

"The sampling efficiency of the MWAC samplers is between 75 % and 90 % for 30 µm dust (Goossens and Offer, 2000), which is within a similar size fraction of the Iwik dust (Friese et al., 2017)."

**Comment Referee #1:**

P7L33-35: As the intensity readings are used and it is calculated with, I would suggest not calling this qualitatively. It does not become clear from the manuscript, whether the procedure of Weltje and Tjallingii (2008) was performed (i.e. log ratios were calculated, relative detection efficiency and specific matrix calibration for the particular setup were obtained), or whether the raw intensity readings were used.

**Reply to comment:**

*We concur with the Reviewer and clarified the section concerning the XRF analyses, in order to avoid confusion. We analyzed homogenized dry sediment trap samples, which largely avoids difficulties typical for XRF scanning on wet sediment cores, such as interference of interstitial water and/or sample roughness effects (as described in Tjallingii et al., 2007 and Weltje and Tjallingii, 2008). Therefore the intensities we measure closely resemble concentration data (i.e. follow linear calibration lines). To elaborate this we extended and adapted the section on XRF scanning of the dust samples. We now show that our XRF intensities closely resemble variability of concentration data, by addition of X-Y plots of concentration data to XRF intensities showing high correlations for our target elements (Appendix S1). Also, we replaced the reference to Weltje and Tjallingii (2008) to that of Tjallingii et al. (2007) to avoid the confusion about data processing, as Tjallingii et al. (2007) already shows that dry samples follow much better concentration data. The measurements we did on our dry dust samples can hence be approached statistically similar as concentration data.*

**Changes made in manuscript:**

We changed the XRF paragraph in the method section:

Page 8, line 1 – 12

"The elemental composition of each sediment trap sample was determined by X-ray fluorescence (XRF) using the Avaatech XRF core scanner (Richter et al., 2006). This analytical technique has the important advantage that it is non-destructive, allowing that very small-size samples – such as sediment trap samples – can be used for other analyses after measurement. XRF scanning results in semi-quantitative compositional data (Richter et al., 2006), being expressed as intensities (i.e. counts or counts per second), which we normalize to the total counts to take into account the closed sum of geochemical data. We analyse our data as normalized element intensities, using the advantage that XRF-scan measurements on homogenised dry sediment trap samples largely avoid physical properties biasing, e.g. wet down-core XRF measurements (Tjallingii et al., 2007;Weltje and Tjallingii, 2008). For dry-powder samples, Tjallingii et al. (2007) showed that element intensities are proportional to their chemical concentration, which we confirm by measuring 13 standards with various matrices, including marine sediments that have a similar matrix to sediment trap samples (Supplement S1)."

**Comment Referee #1:**

P7L34-35: "these two elements": Weltje and Tjallingii suggest a method based on two elements. If you normalize to the total counts here, you do right that, what Weltje and Tjallingii try to avoid by using two different elements. Please consider revising the data processing or write more clearly, if the method of Weltje and Tjallingii is used.

**Reply to comment:**

*As described above (P7L34-35) we clarified the section about XR-scan analyses, elaborating on the method used and making clear that our data closely resembles concentration data.*

**Comment Referee #1:**

P8L25: If there were wind speed measurements available at the MWAC station, it should be considered giving the information together with the horizontal mass flux, as averages or as times, where the wind speed was higher than an appropriate deflation threshold. That way, it could be estimated, whether emission was locally dominated.

**Reply to comment:**

*We now downloaded monthly wind speed data around the MWAC station from Giovanni of the Goddard Earth Sciences Data and Information Services Center (GES DISC) provided by NASA. They show average speeds of around 3 to 7 m/s with higher wind speeds in summer and lower wind speeds in fall. For an accurate deflation threshold, we do not have enough information since the main factor affecting the threshold velocity is the interparticle cohesion force which is dependent, amongst others, on the particle size, density and soil moisture (Pye, 1987). However, according to Pye (1987), the threshold velocities for desert pavements are between 1.5 and 3 m/s. Since the average wind speeds are higher throughout the entire sampling period, we suggest that the captured dust is not only locally dominated and also might come from adjoining areas. For more information, see Friese et al. (2017), in which exactly these processes are described.*

**Changes made in manuscript:**

We added the following sentences to the manuscript:

Page 9, line 10 – 16

"The total mass fluxes of the MWAC samplers at 290 cm height show significant month-to-month differences (Fig. 2). Highest dust fluxes of around 360 and 230 g m$^{-2}$ d$^{-1}$ were found during spring in April 2013 and during summer in August 2013, respectively. Local wind speeds at 10 m above the displacement height (MERRA model, NASA GES DISC) show monthly average velocities between 3 and 7 m s$^{-1}$ with higher wind speeds in early summer and lower wind speeds in fall. Throughout the entire sampling period, the wind speeds are above desert pavement threshold velocities of 1.5 to 3 m s$^{-1}$ (Pye, 1987), suggesting also dust contribution from adjoining areas."

**Comment Referee #1:**

P9L3: This is actually an import.

**Reply to comment:**

*This is an interesting remark, which probably results from the different "languages" that are used within the different scientific disciplines. Depending from which way you look at it, it might be import, yes. However, we talk about the export fluxes from the surface to the deep ocean which are intercepted by the sediment traps. Therefore, we leave it as 'export' since this is the common way in oceanography to express downward particle fluxes.*

**Comment Referee #1:**

P9L15 and P9L38: If there are significant differences between the upper and lower traps recorded: where does the additional mass come from, where does the lost mass go to? The differences are not small.

**Reply to comment:**

*There are differences in the upper and lower traps at site M2 and M4: there is less material caught in the lower trap in comparison to the upper trap at site M4 and there is more material collected by the lower trap in comparison to the upper trap at site M2. While the situation as site M4 shows the 'normal' situation, the situation at site M2 is challenging. The sediment traps are 2300 m apart, over which distance the sinking particles are potentially subject to a variety of processes, such as remineralization, disaggregation, or repacking by zooplankton. This happened at site M4 with the result in lower particle fluxes in the lower trap. However, in addition to the biological processes, advection of water displaces particles on the horizontal plane might take place, resulting in a greater catchment area of the sinking particles (Waniek et al. 2000). We attribute the additional mass in the lower trap at M2 to be the result of this greater catchment area of the lower trap.*

**Changes made in manuscript:**

Page 22, line 2 – 7

"However, the settling pathway might be different, as indicated by the higher total mass fluxes in the lower trap than the upper trap at site M2. Over the 2300 m distance of the upper and lower trap, the sinking particles are potentially subject to a variety of processes including remineralisation, disaggregation, or repacking, as well as

horizontal movement of the particles, resulting in a greater catchment area (Waniek et al. 2000). Therefore, the enrichment in the lower trap at site M2 might result from the greater catchment area of the deeper trap (Siegel and Deuser, 1997;Waniek et al., 2000)."

**Comment Referee #1:**

Fig. 4 is on the small side. In particular the relative mass percent are difficult to compare. I suggest having an additional column of cumulative/stacked plots in white/gray/green/orange instead of the small separate plots. I would also suggest dividing the x axes by seasons instead of 2-month intervals, as seasons are discussed in the text. Moreover, vertical grid lines at the season boundaries would surely help to read particularly the upper plots.

**Reply to comment:**

*We fully agree with the reviewer. The figure is on the small side but like this it is showing all the data on one page. We tested several options and we think that this is the best way to present the data. However, we acknowledge the suggestion of the seasonal grid which makes the traps more comparable. In addition, we provide the entire data set in a table on Pangaea for download.*

**Changes made in manuscript:**

Seasonal grid added in Fig 4. (now 3) and Fig. 5 (now 4).

**Comment Referee #1:**

Fig. 5: What are the diamonds in the modal grain size?

**Reply to comment:**

*The diamonds in figure 5 in the grain size data at site M3 were considered as outliers (van der Does et al. 2016).*

**Changes made in manuscript:**

We added the following clause in the figure caption:

Page 15, line 3 to 5

"Black line represents modal grain sizes of Saharan dust of the same sample, diamonds in M3 are outliers (data Van der Does et al. 2016)."

**Comment Referee #1:**

P13L14: "Saharan dust is characterized by exclusively lithogenic elements": I suggest "We identify Saharan dust by elements", as in particular K in the atmosphere might have different sources (biomass burning), which might be not relevant in your case.

**Reply to comment:**

*Yes, we agree.*

**Changes made in manuscript:**

We adopted the mentioned suggestion in the manuscript:

Page 16, line 15 to 18

"We identify the residual mass fraction as lithogenic fraction by elements such as titanium (Ti), aluminium (Al), iron (Fe) and potassium (K). These elements are incorporated in mineral dust, in especially in aluminosilicates and feldspar, oxides and hydroxides and are incorporated in crystal lattice (Scheuvens et al., 2013)."

**Comment Referee #1:**

P13L19: If the procedure of Weltje and Tjallingii was implemented, the following comment can be neglected, otherwise: Correlating relative readings ('normalized intensities') can produce quickly spurious correlations, in

particular, if one of the major elements shows independent variations (e.g., Ca, Si). I highly suggest not doing that, but investigating elemental ratios instead; e.g., Ti/Al vs. Ca/Si.

**Reply to comment:**

*We would like to demonstrate with the XRF data that the calculated residual mass fraction (TM - CaCO3 - OM - BSiO2) fully includes the lithogenic fraction in the sediment trap samples and is of the same composition as the pure Saharan dust samples at Iwik. This is best shown by plotting the individual lithogenic elements against the residual mass fraction (Fig. 7 (now 6) in manuscript). We explored the suggested ratios as well (see figure below) but we think that this only complicates the visualization since lithogenic (Ti/Al) and both lithogenic and biogenic elements (Ca/Si) are compared to each other. Therefore, we did not change it in the manuscript. Moreover, as we now strengthened our section on the XRF-scan measurements (see comment to P7L33-35), we are convinced that our measurements closely resemble concentration data, and as such a correlation of XRF data to residual mass fraction is warranted.*

[Figure]

**Comment Referee #1:**

Fig. 9: What are the units on the y-axis?

**Reply to comment:**

*Figure 9 (now 8) is showing the biogenic silica measured by sequential leaching (y-axis) versus the total silica measured by XRF (x-axis). Indeed, the y-axis was not properly labelled. It gives the biogenic silica as weight percentages (wt %).*

**Changes made in manuscript:**

We changed the axis name in figure 8 (Page 19):

[Figure]

**Fig 8. Biogenic silica versus total silica in particulate fluxes at five sites across the North Atlantic with sampling site M1 in the east (right) and M5 in the west (left).**

**Comment Referee #1:**

P15L13: What does PSA mean?

**Reply to comment:**

*PSA stands for **p**otential **s**ource **a**rea, it was used in Scheuvens et al. (2013), but was not yet is not clearly addressed in the manuscript.*

**Changes made in manuscript:**

We added the missing term in the manuscript:

Page 19, line 7 – 9

"The land-based dust collectors in Iwik are located in the coastal region of western Mauritania, in potential source area 2 (PSA 2), which is one of the major source areas of dust that is transported across the Atlantic Ocean to the Americas (Scheuvens et al., 2013)."

**Comment Referee #1:**

P15L18: How can the horizontal flux from the MWAC sampler can be compared to the vertical flux of the sediment traps? At least, information on wind direction is necessary here. Also, it seems to me rather improbable that horizontal surface flux on the Northafrican continent should be closely linked to ocean deposition near the Caribbean, as commonly a lot of dust is transported aloft in the Saharan air lay for longer distances (as detailed below in the manuscript).

**Reply to comment:**

*The reviewer is correct; there is a difference between horizontal transport (Iwik) and downward deposition (marine traps) of dust. However, initially we thought that we might see an elevated dust flux in both the MWACs and the sediment traps (at least at site M1) in the same months due to high dust activity on the continent and a subsequent high dust flux in the atmosphere. Unfortunately, this simplification seems unjustified. We only see that the horizontal dust flux in the MWAC sampler is much higher due to its position on the continent. The wind speed is 3 to 7 m/s on monthly average, whereas we do not have information about the direction. Generally, the wind is blowing westwards from the Mauritanian coast which would transport the dust to the direction of site M1. The deposited residual flux at site M1 is due to mass loss much lower.*

**Changes made in manuscript:**

We changed the following paragraph in the manuscript:

Page 19, line 13 – 15, page 20, line 1 – 4

"Overall, the horizontal Saharan dust flux from the land-based MWAC samplers in Iwik cannot be compared directly to the downward flux in residual mass in the sediment traps. However, they are compositionally the same, especially at the ocean site M1 as indicated by their similar Ti-Al slopes (Fig. 7). From site M1 we observe an overall decrease in both the residual mass and the marine biogenic matter fluxes westward to M2/M3 and M4."

**Comment Referee #1:**

P15L20,25: By the notion 'content' in relation with XRF, reference is made to a linear contribution (mass, volume, moles). This seems in contradiction to the method section, where normalized intensities are referred to, which do not reflect directly a mass without proper calibration.

**Reply to comment:**

*The wording is a contradiction; however, we show that the element counts are proportional to the element contents. Nevertheless, due to a missing quantitative calibration, which is beyond the scope of this paper.*

**Changes made in manuscript:**

We changed the following sentences:

Page 20, line 4 – 11

"The much higher counts of lithogenic elements Ti, Al, Fe and K found in the pure dust at Iwik result from the absence of diluting biogenic matter produced in the marine realm. While the Ti-Al slopes are the same at Iwik and ocean sites M1 through M4U, and Al, Fe and K counts are most similar at Iwik and M1 as well, but become significantly higher towards the west (Fig. 7), it suggests a downwind change in mineralogical composition of Saharan dust. At ocean sites M4L and M5L, both the Ti/Al slope is lower and the Al counts are higher especially at M5, indicating that a second source in addition to Saharan dust is involved, which may be derived from admixture of re-suspended clay-sized sediments advected towards the deep sediment traps."

**Comment Referee #1:**

P16L7: Particle size would play the primary role (over shape) in settling speed, so it should be termed like that instead of the ambiguous "lighter". Moreover, the siliciumrich quartz and feldspar particles are usually found at larger particle sizes, which can be extracted from the referenced literature, e.g. the Kandler et al. (2009) paper. As result, this compositional change is consistent with the downwind fining.

**Reply to comment:**

*Indeed, settling speed is dependent on particle size, size and density. The 'lighter' clay particles are, however, not unconditionally also finer. Moreover, they are kept in suspension more easily due to their flat shape.*

**Changes made in manuscript:**

We exchanged 'lighter' with 'finer:

Page 20, line 16 – 20

"Such a downwind decrease in the relative amount of quartz may result from the increase of finer particles due to their slower settling speed, and a relative increase in platy clay minerals such as micas suspended in the atmosphere that can be transported over greater distances."

Korte et al. present an interesting account of the total mass and compositional fluxes of African dust across the Atlantic Ocean. This study is complimentary to the sizedependent transport study by van der Does et al. (2016). Generally, the authors report higher concentrations of mass closest to Africa and the Caribbean, yet the composition changes as dust is transported downwind, indicating different dust sources of the study area. Although the authors present a unique story for ACP, there are several issues that need to be addressed prior to publication in its final form.

**Comment Referee #2:**

General comments:

Previous studies have demonstrated atmospheric transport of North African dust is strongest in the summer (Jun – Aug), which is not consistent with the current work. Can the authors comment on why this inconsistency exists?

Chiapello, I., and C. Moulin (2002) TOMS and METEOSAT satellite records of the variability of Saharan dust transport over the Atlantic during the last two decades (1979– 1997), Geophys. Res. Lett., 29(8), doi:10.1029/2001GL013767.

Perry, K. D., T. A. Cahill, R. A. Eldred, D. D. Dutcher, and T. E. Gill (1997) Longrange transport of North African dust to the eastern United States, J. Geophys. Res., 102(D10), 11225–11238, doi:10.1029/97JD00260.

**Reply to comment:**

*Apart from potential errors in satellite-derived quantifications of dust transport like for example the problems with clouds obscuring the visibility of dust, there are several possible reasons why we do not see such a pronounced seasonality in all our sediment traps or direct dust measurements on land as opposed to satellite imagery. First, we show the residual mass fluxes in which the dust is incorporated. This is the fraction which is calculated by subtraction of the biogenic particles from the total mass, and for now the best guess of dust fluxes when comparing to other sediment trap studies dealing with Saharan dust. At site M1, closest to the African continent, the residual mass fraction does show a clear seasonality with higher fluxes in summer and fall, confirming previous studies showing atmospheric Saharan dust transport strongest in summer (Chiapello et al. 2002). However, this pronounced seasonality at M1 becomes weaker with distance from the source, possibly attributed to changes in conversion factors in areas with less Saharan dust deposition. Another reason might be that there is a time lag between dust entering the surface ocean and dust-particle aggregates arriving at the sediment trap depths. The sediment traps collect all the aggregates (dust plus biological constituents) settling through the water column as soon as they are heavy and dense enough to sink. This may be in the order of a few weeks to a month. However, this would not change an entire season but makes it more challenging to compare atmospheric satellite data to deep ocean particle fluxes.*
*For the biological part, seasonality in the equatorial Atlantic is generally weak due to the small changes in water temperature, sun light and nutrient availability.*

**Changes made in manuscript:**

We added the following sentence to the results at site M1:

Page 10, line 11 – 17

"Consequently, the residual mass fraction shows the opposite, with a low relative abundance in winter and spring and a high in summer and fall. The variation of the residual mass flux is consistent with the long-term seasonal transport of Saharan dust above the eastern tropical North Atlantic (Chiapello and Moulin, 2002). The residual mass flux ranges from 32-71 mg m$^{-2}$ d$^{-1}$ with an average of 47.7 mg m$^{-2}$ d$^{-1}$ (17.4 g m$^{-2}$ a$^{-1}$) and slightly elevated values during summer and fall but with a peak in March (#9). A pronounced seasonality is also seen in the grain-size distributions of the same material with coarser grained Saharan dust in summer and finer grained dust in winter (Van der Does et al. 2016)."

**Comment Referee #2:**

The authors should more clearly link the spatiotemporal heterogeneity in the composition of the current work to the size-dependent depositional trends of van der Does et al. (2016), i.e., the smaller clays are transported farther while the larger quartz are deposited closer to Africa.

**Reply to comment:**

*Yes, we can link the small clay particles in the west at site M5 to the aluminum and potassium, where we see a linear correlation.*

**Changes made in manuscript:**

We added a sentence to the XRF ratios:

Page 16, line 22 – 26

"However, there are spatial gradients from east to west relating to a continuous enrichment of especially Al and Fe from M1 to M5, and to an offset of higher K at M5, while Ti stays constant. The modal particle sizes of the dust (Van der Does et al. 2016) do not show a relation to the lithogenic elements. Best relations found were in the west at site M5 where sizes and lithogenic elements are negatively correlated (Al, $R^2$=0.54 and K, $R^2$=0.44)."

**Comment Referee #2:**

The first paragraph of the introduction contains a nice summary of previous work, but it reads somewhat like a list of dust observation references with reporting their main findings. The first few sentences do contain this information, stating the quantities of dust measured, but what about the latter half of this paragraph? In order to demonstrate how this study is an advancement of and unique from previous work, discussing what these previous studies found is useful. Additionally, the introduction, in general, is lacking broader implications for evaluating Saharan dust transport. Without the broader impacts, the motivation behind this work is not evident.

**Reply to comment:**

*We reorganized the introduction and broke it down to the main findings regarding Saharan dust sampling in the east on land, the atmosphere and in the ocean, as well as Saharan dust sampling at Barbados. The different Saharan dust transport mechanisms are introduced as well, about how they differ in altitude in winter and summer and have different path ways.*

**Changes made in manuscript:**

Introduction, page 2 and 3

[revised manuscript text omitted]

**Comment Referee #2:**

Clarification on some of the methods and calculations is needed, specifically pertaining to biogenic Si, the experimental setup with the mast and moored trap locations, and horizontal flux transport. See specific comments below:

Was BSiO2 measured in the Iwik samples to serve as a control; to truly indicate that this Si was of a marine biogenic origin? How was this silica discernable from the mineral dust Si? What would be helpful is if the methods for extracting and detecting BSi contained more explanation, especially since this is not a technique one commonly encounters in an atmospheric journal. For instance, why the 660 nm laser; does it differentiate biogenic from mineral Si somehow, or is there something done in tandem to the diatom reference material? How does this differ from, say, XRF, in terms of measuring Si? More explanation would be helpful for someone who is not familiar with this technique.

**Reply to comment:**

*The reviewer is correct, it would have been desirable to measure the biogenic silica content in the Iwik samples as well, if only to demonstrate there is no biogenic silica in these samples. However, we were not able to do so, due to insufficient material. For the used method around 100 mg of pure dust would be needed, is more than we have in the MWAC samples from Iwik.*

*The method to determine biogenic silica is well-accepted and much applied in marine sciences and is based on an alkaline (NaOH) leaching technique dissolving the amorphous silica (opal) in the presence of lithogenic silica minerals (Koning et al. 2002). Marine sediments surely consist of a number of silica fractions; of biogenic origin, clay minerals, aluminosilicates and quartz, from which the quartz and aluminosilicate minerals are virtually insoluble within the time span of a normal leaching experiment (60-90 min). For clay minerals it was shown that they dissolve at a constant, much slower rate and independently from the biogenic silica. Extrapolation of the linear 'clay dissolution line' to time zero would correct for the contribution of non-biogenic silica resulting in the $BSiO_2$ content of the sample (DeMaster 1981). The determination of dissolved silicon is based on the formation of a yellow silicomolybdate acid when an acid sample is treated with the molybdate solution (Grasshoff et al. 1983). The chemicals added for the used method are a sulphuric acid – molybdate solution, acidifying the 0.5 M NaOH to pH 2, allowing to form the molybdate complex, followed by oxalic acid, avoiding reduction of excess molybdate and ascorbic acid as a reductant to stabilize the blue complex. Eventually, the absorbance of the blue complex is measured at 660 nm in the spectrophotometer, which is the defined wavelength where no dilution is necessary (Grasshoff et al. 1983). As described above, the leaching technique is only measuring the biogenic silica while the XRF scanning determines the total silica fractions, including all the biogenic silica, clay minerals, aluminosilicates and quartz. The principle of the XRF analysis is based on excitation of electrons by incident X-radiation (Weltje & Tjallingii 2008). It is a semi-quantitative method which results are presented in form of element intensities (counts) and/or ratios of these element intensities. By contrasting both methods we showed the differences in composition with decreasing contribution of lithogenic silica towards the west to the extent that almost all silica is of biogenic origin in the west at site M5.*

**Changes made in manuscript:**

We changed the method description as follows:

Page 7, line 19 – 26

"A standard amount of 25 – 30 mg of ground sample was placed in a 0.5 M NaOH solution at 85° C to dissolve the biogenic silica, which was subsequently reacting with a sulphuric acid-molybdate solution to form a blue molybdate complex. The complex was prevented from molybdate reduction and stabilized by adding oxalic and ascorbic acid, respectively. The solution was flushed through a photocell where the absorption of the blue complex was measured at the defined 660 nm (Grasshoff et al., 1983) and recorded every second. Each sample was run for 60 to 90 minutes. Results were evaluated with a weekly measured standard calibration curve ($R^2 > 0.99$) and calculated with the MS Excel data solver tool, extrapolating the dissolution curve to time zero to correct for contribution of non-biogenic silica (DeMaster, 1981)."

**Comment Referee #2:**

Horizontal flux transport indicates transport over a set distance, however, the authors present this factor as g m2 d-1, which indicates a time-dependent flux. I would assume both masts would be used to calculate the true horizontal flux of dust. Perhaps the wording is throwing me off here, but it seems as if this calculation is at one MWAC sampler as indicated on p8, l11, yet there were 20 samplers (2 masts with 5 pairs of samplers each). How far apart were these masts? Additionally, can the authors comment on how much of the dust flux is missed at higher altitudes, since the samplers extend up to 2.9 m but dust transport can happen up to 5 km?

**Reply to comment:**

*The main interest of this study was to sample the local dust within the (accessible) source areas, which we can chemically compare to the dust we find in the sediment traps. The two dust masts are less than 30 m apart and hold 20 samplers in total. The given flux data is, however, only from the samplers at 290 cm height, in order to avoid sediment input by saltation. How much dust is missed on its way into the high atmosphere is difficult to estimate, and beyond the scope of our study. The amount is mainly dependent on the particle size and shape, keeping the dust particles in air suspension and allowing them to be uplifted to higher altitudes. We tried to*

*translate the amount of 182 Tg a$^{-1}$ dust leaving the coast of North African given by Yu et al. (2015) to horizontal dust fluxes per m$^2$. This simple conversion results in a dust flux of a few tens g m$^{-2}$ d$^{-1}$, assuming a homogeneous horizontal and vertical dust distribution (over 10° S to 30° N and 4 km or 7 km height) which is, however, impossible to compare with the horizontal mass flux in Iwik.*

**No changes made in manuscript.**

**Comment Referee #2:**

For those of us not familiar with sediment traps, why were they located so deep and not closer to the surface where the dust would initially deposit and where the Northern Equatorial Current would carry dust from east to west? It seems like there would be a lot of room (literally) for different processes to remove the dust during sedimentation, like the deeper circulation from the conveyor belt in that region (N-to-S directionality). Since many readers will not have an oceanographic background, but are well versed in atmospheric transport mechanisms and how sensitive aerosol transport can be due to shifts in large and small scale circulation patterns, it would be helpful to provide a few sentences early on regarding Atlantic Ocean circulation and how this could potentially impact samples collected 2-3 km deep where the traps were located.

**Reply to comment:**

*Sediment traps are a handy tool for direct and accurate time-series measurements of settling particles in the ocean when conditions are favorable (Knauer & Asper, 1989). Among these conditions are that current velocities are < 12 cm/s, no deep eddy penetration and a stable vertical mooring cable, no biofouling or fisheries activities. These conditions are only met in the deep ocean. When sediment traps are too close to the ocean's surface, they suffer from surface currents suddenly changing the trap position, resulting in an unreliable particle flux. Around 99% of the material in the ocean is decomposed and recycled within the euphotic zone, only 1% of the produced particles escape to the deep ocean, which eventually account for the sequestered carbon. However, as the reviewer states, there is a lot of space for lateral transport affecting the total particle flux. Waniek et al. (2000) suggest that particles are moving with an angle of about 1° relative to the surface with a typical horizontal velocity in the range of 10-20 cm/s and a particle sinking speed of 100 – 200 m/d. Therefore, it might be possible that the lower trap collects more or less material than the upper trap, depending on the pathway of the material. The best we can do is to obtain a thorough understanding of the mooring line motion using the current- and tilt meters attached to the moorings to evaluate the performance of the sediment traps. This is what we did and explained in the method section:*

**Changes made in manuscript:**

Page 4, line 12 – 14

"Sediment traps are a common tool for direct and accurate time-series measurements of settling particles in the ocean when conditions are favourable, e.g. low currents (< 12m s$^{-1}$), no deep eddy penetration and a vertical mooring line (Knauer and Asper, 1989)."

**Comment Referee #2:**

For the calculations of carbonates and OM: Where does the 8.33 correction factor come from for CaCO3? For OM, it seems like this calculation is likely underrepresenting total OM. Why isn't the TON included in OM? There is more than just C in organic matter (O, S, H, N to name a few; see references below as a couple examples), so if these aren't accounted for, the authors should clearly state the limitations and caveats with this calculation. And lastly, what is Corg and how was this measured/calculated (is it just TOC)?

Hazem S El-Zanan, Barbara Zielinska, Lynn R Mazzoleni & D. Alan Hansen (2009) Analytical Determination of the Aerosol Organic Mass-to-Organic Carbon Ratio, Journal of the Air & Waste Management Association, 59:1, 58-69, DOI: 10.3155/1047- 3289.59.1.58.

L. M. Russell (2003) Aerosol Organic-Mass-to-Organic-Carbon Ratio Measurements. Environmental Science & Technology 37, 2982-2987.

**Reply to comment:**

*The calculations used are the common way to measure and calculate deep ocean fluxes, which is a well-accepted and much applied technique in marine sciences, including for sediment trap samples. The total carbon (TC) and total organic carbon (TOC) were analyzed with an elemental analyzer. From these two analyses the total inorganic carbon (TIC) is calculated by subtracting the organic carbon from the total carbon (TC – TOC = TIC). TIC contains all calcium carbonate ($CaCO_3$), which is also the most concrete factor in this calculation. It is calculated as $CaCO_3$ = TIC \* 8.33 based on the molecular weight of $CaCO_3$ (Ca (40) + C (12) + O (16 \*3) = 100), divided by the carbon measured (C = 12) resulting in 100/12 = 8.33. For determination of the organic matter (OM), conversion factors vary since its molecular composition is highly heterogeneous and hence poorly constrained. Indeed, the reviewer names additional elements incorporated in organic matter besides organic carbon. Depending on which molecular structure one uses (OM simply as $CH_2O$, or the more sophisticated $(CH_2O)_{106} (NH_3)_{16} H_3PO_4$ derived from the global Redfield ratio), one ends up with different calculation factors. However, the most commonly used factor of 2 is chosen here for better comparison to other studies (Wefer & Fischer, 1993, Fischer et al. 2016, Neuer et al 2002) dealing with Saharan dust fluxes. $C_{org}$ is the same as the TOC.*

**Changes made in manuscript:**

We adapted the method section on the carbon analysis as follows:

Page 7, lines 11 – 17

"Carbonates were calculated as $CaCO_3$ = (TC – TOC) \* 8.33 and organic matter as OM = 2\*TOC. $CaCO_3$ is calculated as (TC – TOC) \* 8.33, and organic matter as OM = 2\*TOC. The conversion factor for $CaCO_3$ is based on its stoichiometry, given 100 mol/g of $CaCO_3$ for 12 mol/g of carbon, resulting in a factor of 100/12 = 8.33. The conversion factor for organic matter varies from 2.0 in the eastern Atlantic Ocean to 2.5 in upwelling areas (Jickells et al., 1998;Klaas and Archer, 2002;Thunell et al., 2007;Fischer and Karakas, 2009) due to poorly constrained composition of the actual organic matter. We chose to use the factor of 2 for better comparison to particle fluxes influenced by Saharan dust deposition off Cape Blanc (Fischer and Karakas, 2009;Fischer et al., 2007;Wefer and Fischer, 1993)."

**Comment Referee #2:**

Some clarification on the elemental components used for mineral classification is needed. For determining the elemental compositions of lithogenic and biogenic minerals, why is Ca not used? Wouldn't this corroborate the carbonates calculation from TOC and TC? For K, was biomass burning K removed? Central Africa is prone to fires and thus a large source of biomass burning aerosol (K is commonly used as a biomass burning marker). This could simply be resolved by calculating soil K, where [non-soil K] = [K] – 0.6[Fe] and [soil K] = [total K] – [non-soil K] from Kreidenweis et al. (2001).

Kreidenweis, S. M., Remer, L. A., Bruintjes, R., and Dubovik, O. (2001) Smoke aerosol from biomass burning in Mexico: Hygroscopic smoke optical model, J Geophys Res-Atmos, 106, 4831-4844.

Along these lines, what is the possible contribution from sea salt minerals (i.e., K) recrystallizing during sample processing to the elemental concentrations? Even though the samples were washed for salts, could the authors comment on how salts could potentially recrystallize onto insoluble particles during processing?

**Reply to comment:**

*We used the XRF measurements to evaluate the lithogenic fraction in the sediment traps. We did not use Ca since in our samples it is almost exclusively biogenic in nature, formed by carbonate producing phyto- and zooplankton like coccolithophores and foraminifera. Consequently, also the Ca as measured by XRF analysis is positively correlated with the TIC measured in the sediment traps ($R^2$=0.73).*

*Indeed, K is used as a biomass burning marker. Our analysis does not distinguish between K-bearing lithogenic minerals (feldspars, clay minerals in the residual mass fraction) and K-bearing black carbon from biomass burning among the marine organic matter. All sediment trap samples are kept in their original liquid solution prior to processing and then washed three times with Milli-Q water prior to analysis, to remove any sea salts prior to drying. If there would be any sea salts left, it would be the same for every sample and would bias the samples equally. Given the high correlations between K, Ti and Al, the K would not be derived from any sea*

*salts since seawater concentrations of Al and Ti are in the nano-picomolar range, which also argues that any K contribution from black carbon associated with biomass burning is minor.*

**Changes in manuscript:**

Page 16, lines 15 – 22

"We identify the residual mass fraction as lithogenic fraction by elements such as titanium (Ti), aluminium (Al), iron (Fe) and potassium (K). These elements are incorporated in mineral dust, in especially in aluminosilicates and feldspar, oxides and hydroxides and are incorporated in crystal lattice (Scheuvens et al., 2013). Although K is also detected in biomass burning aerosols (Cachier et al. 1995), the high correlations between K and the other lithogenic elements show that the contribution of biomass burning K is minor in our samples. Normalized intensities of the lithogenic elements are highly correlated with the residual mass fraction and are therefore thought to represent the deposition flux of Saharan dust (Fig. 6)."

**Comment Referee #2:**

None of the figures show uncertainties/error bars. For instance, the methods section says the total mass fluxes (MWAC) have a 90% collection efficiency, so that remaining mass not collected should be accounted for in the measurement uncertainty. In order to clearly draw statistically significant relationships between the months, error bars should be provided.

**Reply to comment:**

*Yes, the reviewer is right, errors should be accounted for. In general, the MWAC samplers are a good and widely-accepted tool to trap wind-blown sediments. However, the efficiency ranges have been demonstrated to vary. They are > 75 % and usually > 90 % according to Goossens and Offer (2000). For the revised manuscript we chose the minimum and maximum deviations, i. e. 75 and 90 %, and added those to the graph, in addition to wind speed data (Fig. 3 (now Fig. 2)). However, we would like to stress that the horizontal dust fluxes are only shown as a side information. The main purpose of these samples is the elemental composition, to which we want to compare to the sediment trap samples. For detailed information on the MWAC samples we would like to refer to Friese et al. (2017).*

*The errors for the marine particle fluxes are only given in the methods section. We are aware that they add up for the residual mass fraction since this is the fraction which is calculated from the remainder of the total to the biogenic particles. We do not show them in the figures, since it would make them unreadable and as this method is well-established (Wefer and Fischer, 1993;Neuer et al. 2002;Fischer et al. 2007), and most other literature only discusses average fluxes, we chose to leave them out from the figures as well.*

**Changes made in manuscript:**

[Figure]

**Figure. 2. Horizontal transport fluxes of Saharan dust from the land-based MWAC sampler in Iwik, Mauretania (19° N, 16° W) between January and December 2013 in orange bars. Error bars show the MWAC sampler efficiency of 75 to 90 %. Monthly averaged wind speed at 10 m above displacement height (MERRA model) around the Iwik location is indicated as a black line (data obtained from the Giovanni online data system, NASA GES DISC). AGL = above ground level**

**Comment Referee #2:**

The seasonal maxima of the MWAC versus moored traps are offset. Can the authors comment on this in the discussion section?

**Reply to comment:**

*The MWAC sampler and the sediment traps are two different kind of traps. The MWAC sampler is a really local aerosol sampler on land, collecting all indigenous transported dust events, while the marine sediment traps collect deposited dust. In addition, they do not sample at the same time intervals, which makes it even more difficult to compare. However, and that is what we would like to show here, the dust collected in both the MWAC aerosol samplers and the deep-ocean sediment traps have the same chemical composition, although they are so far apart from each other and sample a different process (transport versus deposition). A detailed description of the working of the MWAC samplers is beyond the scope of this paper and details are provided in Friese et al. (2017), ACPD.*

**Changes made in manuscript:**

We adapted the following section in the discussion:

Page 19, line 13 – 15, page 20, line 1 – 4

"Overall, the horizontal Saharan dust flux from the land-based MWAC samplers in Iwik cannot be compared directly to the downward flux in residual mass in the sediment traps. However, they are compositionally the same, especially at the ocean site M1 as indicated by their similar Ti-Al slopes (Fig. 7). From site M1 we observe an overall decrease in both the residual mass and the marine biogenic matter fluxes westward to M2/M3 and M4."

**Comment Referee #2:**

Specific comments:

P1, l17: The way this is written, it sounds like the authors are stating that the African coast is the source. Perhaps change to "African continent" or "North Africa".

**Reply to comment:**

*We agree on the comment.*

**Changes made in manuscript:**

We changed the phrasing on page 1, line 19:

"Massive amounts of Saharan dust are blown from the coast of northern Africa across the Atlantic Ocean towards the Americas each year."

**Comment Referee #2:**

P6, l8: To be clear up front, it would be helpful to provide a statement saying which analyses were conducted on the MWAC and which were conducted on the moored trap samples.

**Reply to comment:**

*The reviewer is right, a statement on which analyses was applied to which samples is helpful and makes it easier for the reader to follow.*

**Changes made in manuscript:**

We added the following paragraph at the beginning of the methods section:

Page 6, line 10 – 15

"The following paragraphs describe the different methods used to analyse particle fluxes and the element and biological composition of the specific samples. Due to the different nature and amount of material, i.e. the dry dust from the on-land dust collector in Iwik and the wet sediment trap samples from the deep ocean, not all methods were applicable to all samples. While for the marine sediments the total, biological and residual mass fluxes were determined, as well as their element composition, the Iwik samples were analysed exclusively for their total mass and horizontal dust flux at 290 cm as well as their element composition."

**Comment Referee #2:**

P8, l2: State that lithogenic minerals are clays, quartz, and feldspars here. These are in fact listed but much later on.

**Reply to comment:**

*We agree with the reviewer that the lithogenic minerals should be stated earlier.*

**Changes made in manuscript:**

We rephrased the following sentence:

Page 8, line 20 – 24

"The elements Ti, Al, Fe and K were chosen and shown, since they are only present as main or minor elements in lithogenic minerals like clays, quartz, and feldspars, rather than in any biogenic mineral formed in the ocean, while Si represents both lithogenic Si minerals (clays, quartz, and feldspars) but also biogenic produced silica ($BSiO_2$), as found in phytoplankton diatoms or zooplankton radiolarians."

**Comment Referee #2:**

P8, l6: Please provide the chemical compositions/components of these standards.

**Reply to comment:**

*We recognized that the XRF-method needed more clarification. Therefore, we adopted the method section and provided the chemical composition of the XRF-standards in the appendix (S1).*

**Changes made in manuscript:**

Page 8, line 1 – 12

"The elemental composition of each sediment trap sample was determined by X-ray fluorescence (XRF) using the Avaatech XRF core scanner (Richter et al., 2006). This analytical technique has the important advantage that it is non-destructive, allowing that very small-size samples – such as sediment trap samples – can be used for other analyses after measurement. XRF scanning results in semi-quantitative compositional data (Richter et al., 2006), being expressed as intensities (i.e. counts or counts per second), which we normalize to the total counts to take into account the closed sum of geochemical data. We analyse our data as normalized element intensities, using the advantage that XRF-scan measurements on homogenised dry sediment trap samples largely avoid physical properties biasing, e.g. wet down-core XRF measurements (Tjallingii et al., 2007;Weltje and Tjallingii, 2008). For dry-powder samples, Tjallingii et al. (2007) showed that element intensities are proportional to their chemical concentration, which we confirm by measuring 13 standards with various matrices, including marine sediments that have a similar matrix to sediment trap samples (Supplement S1)."

**Comment Referee #2:**

P8, l11-23: The addition of all the figures in parentheses in this paragraph place the figures out of order and cause confusion as they are not fully described (my eye was jumping back and forth between figures and text without enough clarification). It would be easier if references to these figures were eliminated here and remain discussed in the following sections. Additionally, this is a nice summary of what is to come, but perhaps providing a sentence at the end describing how all of these different observations will be used in tandem to develop the main picture "'X" would clearly tie it all together.

**Reply to comment:**

*We agree on the reviewer's opinion that the reader have to jump back and forth between the figures and text.*

**Changes made in manuscript:**

We accordingly adjusted the phrasing of the introductory paragraph of the results:

Page 8, line 34 – 41, page 9, line 1 – 8

"Horizontal transport fluxes from the land-based MWAC sampler are given in g m$^{-2}$ d$^{-1}$. The vertical (downward) deposition fluxes from seven sediment traps deployed across the Atlantic Ocean are treated in downwind succession from east to west, starting at ocean site M1 closest to Africa, to ocean site M5 closest to the Caribbean. For each trap, the relative contribution of the biogenic components CaCO$_3$, BSiO$_2$, OM, and the residual mass fraction are given, in addition to the total mass flux, the flux of each biogenic component and that of the residual mass fraction, expressed both in mg m$^{-2}$ d$^{-1}$ and g m$^{-2}$ a$^{-1}$. The biogenic components are produced by autotrophic phytoplankton and heterotrophic zooplankton. In particular, phytoplankton CaCO$_3$ producers are mainly coccolithophores while zooplankton CaCO$_3$ is mainly from foraminifera and gastropod shells. The BSiO$_2$ is primarily produced by the phytoplankton diatoms and zooplankton radiolarians. Seasonal variations are shown for specific fluxes as deviation from their annual mean, together with the grain-size distribution of Saharan dust (Van der Does et al. 2016) for the residual mass fraction in the same material. To evaluate the residual mass fraction in the sediment traps, we compare XRF data from MWAC samplers with those from the sediment traps. The horizontal mass fluxes of the land-based dust collectors in Iwik are almost 100 % pure dust with negligible contribution of organic matter and fresh water diatoms from paleolakes and can therefore be used for chemical element comparison to the residual mass fraction from the sediment traps. Deviations between the chemical composition of the MWAC samplers and the sediment traps reveal compositional changes."

**Comment Referee #2:**

P8, l26-27: Only Apr and Aug have the spikes (remove Mar and Jul). Fig 3.: Swap the y and x axes to align with how time series are shown in the rest of the figures.

**Reply to comment:**

*We acknowledge the comment of the reviewer.*

**Changes made in manuscript:**

We adjusted Figure 3 (now Figure 2) and added local wind data:

Page 9, line 10 – 16

"The total mass fluxes of the MWAC samples at 290 cm height show significant month-to-month differences (Fig. 2). Highest dust fluxes of around 360 and 230 g m$^{-2}$ d$^{-1}$ were found during spring in April 2013 and during summer in August 2013, respectively. Local wind speeds at 10 m above the displacement height (MERRA model, NASA GES DISC) show monthly average velocities between 3 and 7 m s$^{-1}$ with higher wind speeds in early summer and lower wind speeds in fall. Throughout the entire sampling period, the wind speeds are above desert pavement threshold velocities of 1.5 to 3 m s$^{-1}$ (Pye, 1987), suggesting also dust contribution from adjoining areas."

**Comment Referee #2:**

P9, l16: Why isn't the lower trap shown? At the very least, provide it in the supporting information.

**Reply to comment:**

*We are not sure what the reviewer means here. The text on mass fluxes at ocean site M2 (4.2.2) describes both the upper and lower trap, and so does the figure 4 (now 3) on page 11.*

**Comment Referee #2:**

P13, l14-15: It can also contain Ca in the form of calcite (see Glaccum and Prospero (1980)), although at < 10%. A reference for this statement is needed.

**Reply to comment:**

*Indeed, dust may contain Ca. However, the marine samples are dominated by biogenic Ca produced by calcifying plankton to the extent that anyl lithogenic Ca is heavily overprinted. Therefore, we used the lithogenic elements Ti, Al, Fe and K, rather than the Ca data for the evaluation of Saharan dust. XRD analysis show that lithogenic carbonates account for only 1.3 wt % in the Iwik samples (Friese et al. 2017). Nevertheless, we rephrased the sentence:*

**Changes made in manuscript:**

Page 16, lines 15 to 16

“We identify the residual mass fraction as lithogenic fraction by elements such as titanium (Ti), aluminium (Al), iron (Fe) and potassium (K).”

**Comment Referee #2:**

P15, l11: First mention of DUSTTRAFFIC; should be introduced in the methods section.

**Reply to comment:**

*We agree and removed it entirely.*

**Changes made in manuscript:**

Page 19, line 6 & 7

 “We provide the first comprehensive time series of biogenic particle fluxes and Saharan dust deposition from source to sink across the equatorial North Atlantic Ocean.”

**Comment Referee #3:**

Thank you for your invitation to review the manuscript submitted to the ACP journal by Laura Korte et al. This paper reports on an ambitious sediment trap experiment carried out in the Tropical North Atlantic Ocean. This experiment, which was set up in the frame of the "DUST TRAFFIC" (ERC) project, aims at documenting Saharan dust inputs to the Atlantic Ocean from the West African coast all the way to the Caribbean, along the main path for Saharan dust atmospheric transport. Korte's paper presents an impressive amount of data obtained during the first year of this experiment, including particle flux and composition temporal variability obtained from 7 sediment traps deployed at about 1100-1200m or 3400-3500m depths, at 5 mooring locations. The manuscript also includes data obtained on land in a dust source region in Mauritania. This study nicely complements earlier investigations carried out off West Africa and in the Sargasso Sea and provides, for the first time, simultaneous flux and compositional data across the entire Atlantic ocean. This is, I find, the main originality and interest of this report. In any case, this study represents a remarkable logistical and analytical effort, which is to be commended. One of the interesting outcome of this study is that there appears to be no marked seasonal variability in the downward particle fluxes across the entire Tropical Atlantic (a moderate seasonal variability is apparent on the edges of the transect though, near western and eastern margins, especially in the terrigneous fraction and the biogenic silica, respectively). This study also demonstrates the prominent influence of Saharan dust inputs in the "residual" (i.e., non-biogenic) material along most of the E-W transect (based on a few major element ratios (Al, Fe, K) that compare well with ratios measured in the dust they collected in Mauritania). "Dust" major element composition, still, displays a marked evolution across the ocean, enabling Laura Korte et al. to track the progressive mineralogical (and grain-size) sorting during transport (with the gradual reduction of coarse quartz particles contribution from East to West, as indicated for instance by the non-biogenic Si content). Korte et al. study finally reveals the influence of an additional terrigenous source in the Western Atlantic: major compositional differences are recorded in particles collected near the Caribbean -higher K contents for instance- suggesting a possible influence of fluvial discharges (Amazon and/or Orinoco) in the area. Overall, I find that the paper presents a large number of rare, high quality sediment trap data that will be of great interest to those interested in the Saharan dust transport (including modelers) and its impact on the ocean (and Amazonian) biogeochemistry. I therefore recommend publication after the authors have addressed the minor comments listed below.

**Reply to comment:**

*This reviewer clearly has an oceanic background and is also interested in the oceanic consequences of dust deposition and is very aware of all the implications that are related to the marine organic fractions that are also collected by our sediment traps. However, given that the average ACP reader likely has an atmospheric background, in this paper we focus on the terrigenous sediment fraction rather than all the other marine organic fractions. We think that discussing all the marine organic fractions and their potential implications would make this manuscript unreadable and very long. Rather, we would like to draw the attention of the atmospheric community to our work, in an attempt to bridge the gaps that exist between the different disciplines. We believe that bringing them together will be of benefit for all disciplines. As a result, we try to highlight the Saharan-dust part of our studies and limit the oceanic part as much as possible, in order provide focus for the manuscript.*

**Comment Referee #3:**

Detailed comments

page 2, line 24-25: Very few studies actually measured "fluxes" (in most cases, dust was collected by air filtration), so you may want to rephrase.

**Reply to comment:**

*We reorganized the introduction in general and included your following suggestions.*

**Changes made in manuscript:**

We changed the 'fluxes' into concentrations:

Page 2, line 19 & 20

"All observations of Saharan dust showed a strong seasonality with higher dust concentrations during the winter season close to the dust sources in the east, and higher dust concentrations during the summer season in the Caribbean."

**Comment Referee #3:**

page 2, line 31: Sarthein et al. [1981] (which discusses paleo-dust records) might not be the best reference here – Chiapello et al., [1995] for example, which documents present-day seasonal Saharan dust transport in the trade wind levels, would be more suitable.

**Reply to comment:**

*We thank the reviewer for the additional reference.*

**Changes made in manuscript:**

We exchanged the reference regarding the Saharan dust transport:

Page 2, line 25 – 29

"In winter, easterly winds transport the dust in surface winds at altitudes below three km (Chiapello et al., 1995), when the ITCZ reaches its most southern position, and the dust crosses the Atlantic Ocean in the direction of South America (Prospero et al., 2014;1981)."

**Comment Referee #3:**

pages 2-3, bridging paragraph: Although there is no doubt that the Bremen group has built the longest sediment trap record in this region, providing one of the most valuable trap data time series ever collected –and should be praised for this!– other groups, in the UK and France for instance, have also contributed to ample sediment flux studies in the area (both off West Africa and in the Sargasso sea) within the 90's BOFS and JGOFS programs among other frameworks. Some results obtained in these experiments are actually fully relevant to Korte's study (see below). So, I think it would be useful to provide a more exhaustive list of earlier experiments (at least provide the references).

**Reply to comment:**

*We reorganized the introduction and adapted the bridging paragraph.*

**Changes made in manuscript:**

Page 2, line 32 – 40, page 3, line 1 & 2

"The knowledge of the fate of Saharan dust in between the sources and sinks is, however, limited due to the vastness of the North Atlantic Ocean, though observed by remote sensing (e. g. Liu et al., 2008;Yu et al., 2015;Huang et al., 2010) and shipboard lidar measurements (Kanitz et al. 2014). From the latest approximation (Yu et al., 2015) it can be derived that around 50 Tg a$^{-1}$ of dust is deposited into the eastern equatorial North Atlantic Ocean, and 140 Tg a$^{-1}$ of dust is deposited into the equatorial North Atlantic Ocean and the Caribbean Sea as well as onto parts of the Amazon rainforest. The deposited dust onto the ocean has a great influence on the particle fluxes as well (Jickells et al., 2005;Ittekkot et al., 1992;Armstrong et al., 2009). Sediment trap studies within for example the Joint Global Ocean Flux Study (JGOFS) and Biogeochemical Ocean Flux Study (BOFS), dealing with deep ocean particle fluxes, show elevated total mass fluxes with a high contribution of lithogenic particle fluxes in the North Atlantic Ocean off Mauritania (Jickells et al. 1996). In addition, Saharan dust particles are thought to contribute to the total flux in the Sargasso Sea (Deuser et al., 1988)."

**Comment Referee #3:**

page 3, line 12-14: I find this sentence somewhat unclear (the "other factors" have implication on grain size? on the dust deposition timing?) – please clarify.

**Reply to comment:**

*Friese et al 2016 showed that grain size is not only dependent on precipitation (as suggested by Fischer et al., 2016) but also on wind speed, especially in winter, and on episodic dust storms which lead to anomalously coarse dust particles in the sediment traps off Cape Blanc.*

*This statement is however irrelevant in the reorganized introduction, since we do not mention it anymore.*

**Comment Referee #3:**

Page 3, line 24: "Composition" is a little vague (you may want to specify here what has been characterized in your study: biogenic components (CaCO3, BsiO2, OM. . .), major elements in the residual (i.e., lithogenic) fraction (Si, Al, Fe. . .) etc.).

**Reply to comment:**

*The 'composition' was referring to the dust sampled on Barbados and measured by ICP-MS (Trapp et al., 2010) and was not referring to our particle flux in this study.*

*However, in the reorganized introduction this observation does not occur anymore.*

**Comment Referee #3:**

page 6, lines 20-24: You may want to indicate that sediment traps are supposedly collecting the downward (i.e., "vertical") flux, especially since you just mentioned (line 10) that your land-based collector allows to estimate the "horizontal" dust flux.

**Reply to comment:**

*Indeed, we should clarify that there is a difference in horizontal and vertical flux.*

**Changes made in manuscript:**

We added the suggested vertical flux in the text:

Page 6, line 27 – 29

"For the vertical (downward) marine particle fluxes from the sediment traps, the samples were wet-sieved over a 1 mm mesh, wet-split in five aliquot subsamples using a rotary splitter (WSD-10, McLane Laboratories), washed to remove the $HgCl_2$ and salts, and centrifuged."

**Comment Referee #3:**

page 7, line 34: Which "two" elements? Any? (in this case, you may want to remove "these" or rephrase).

**Reply to comment:**

*We realized that the XRF method needed a bit more clarification and now spend a few more words on the XRF method, including standard reference material in the supplement:*

**Changes made in manuscript:**

Page 8, line 1 – 12

"The elemental composition of each sediment trap sample was determined by X-ray fluorescence (XRF) using the Avaatech XRF core scanner (Richter et al., 2006). This analytical technique has the important advantage that it is non-destructive, allowing that very small-size samples – such as sediment trap samples – can be used for other analyses after measurement. XRF scanning results in semi-quantitative compositional data (Richter et al., 2006), being expressed as intensities (i.e. counts or counts per second), which we normalize to the total counts to

take into account the closed sum of geochemical data. We analyse our data as normalized element intensities, using the advantage that XRF-scan measurements on homogenised dry sediment trap samples largely avoid physical properties biasing, e.g. wet down-core XRF measurements (Tjallingii et al., 2007;Weltje and Tjallingii, 2008). For dry-powder samples, Tjallingii et al. (2007) showed that element intensities are proportional to their chemical concentration, which we confirm by measuring 13 standards with various matrices, including marine sediments that have a similar matrix to sediment trap samples (Supplement S1)."

**Comment Referee #3:**

page 8, line15: I find "daily averages" somewhat misleading (as it suggest that flux was sample with a daily or even sub-daily resolution); "daily mean" (and annual mean) might be less ambiguous; you may also simplify (e.g., ". . .each constituent, expressed both in mg.m-2.d-1 and g.m-2.a-1").

**Reply to comment:**

*We thank the reviewer for the advice and simplified the expression accordingly:*

**Changes made in manuscript:**

Page 8, line 37 – 39

"For each trap, the relative contribution of the biogenic components $CaCO_3$, $BSiO_2$, OM, and the residual mass fraction are given, in addition to the total mass flux, the flux of each biogenic component and that of the residual mass fraction, expressed both in mg m$^{-2}$ d$^{-1}$ and g m$^{-2}$ a$^{-1}$."

**Comment Referee #3:**

4.2.1 to 4.2.5 Mean annual fluxes ("annual averages") for each component of the settling particulate matter (CaCO3, BSiO2, OM, and for the residual fraction) are of great interest too, and should also be provided for each trap (either in the text or in a table). I find such data are clearly missing in the manuscript. These would also allow discussing how they compare with results obtained at nearby locations in earlier studies (e.g., Bory and Newton, 2000; Bory et al., 2001), which may provide useful clues on the spatial (and/or interannual) variability in the Saharan dust deposition in the tropical Atlantic off West Africa.

**Reply to comment:**

*Yes, we agree with the reviewer that these are of great interest, too. However, we wanted to focus on the residual mass fraction since this material includes all Saharan dust and is of most interest for this journal and audience. Providing the averages for all parameters for all traps would be too much detail. In addition, they will be the subject of a parallel manuscript. Still, we added a table with the yearly mass fluxes of the measured parameters showing the decreasing trend from east to west (M1 to M5) in numbers. In addition, these data set will be available from the Pangaea database after final publication.*

**Changes made in manuscript:**

We added a table with yearly mass fluxes of all measured flux parameters.

Page 11, line 28

Table 3. Yearly mass fluxes of measured flux parameters.

| Trap | Total Mass Flux | CaCO₃ | BSiO₂ | OM | Residual mass | CaCO₃ | BSiO₂ | OM | Residual mass |
|---|---|---|---|---|---|---|---|---|---|
| | mg m$^{-2}$ d$^{-1}$ (g m-2 a-1) | % | | | | mg m$^{-2}$ d$^{-1}$ (g m$^{-2}$ a$^{-1}$) | | | |
| M1 U | 111.6 (40.7) | 33.9 | 9.7 | 13.6 | 42.7 | 37.8 (13.8) | 10.9 (4.0) | 15.2 (5.6) | 47.7 (17.40) |
| M2 U | 39.5 (14.4) | 56.0 | 5.9 | 11.3 | 26.8 | 22.2 (8.1) | 2.3 (0.8) | 4.5 (1.6) | 10.6 (3.9) |

| | | | | | | | | | |
|---|---|---|---|---|---|---|---|---|---|
| M2 L | 54.6 (19.9) | 59.2 | 5.7 | 7.2 | 27.9 | 32.3 (11.8) | 3.1 (1.1) | 3.9 (1.4) | 15.3 (5.6) |
| M3 L | 70.4 (25.7) | 52.0 | 6.9 | 8.4 | 32.7 | 36.6 (13.4) | 4.9 (1.8) | 5.9 (2.2) | 23.0 (8.4) |
| M4 U | 84.8 (30.9) | 48.3 | 10.8 | 12.6 | 28.3 | 41.0 (14.7) | 9.2 (3.4) | 10.7 (3.9) | 24.0 (8.7) |
| M4 L | 63.6 (23.2) | 53.0 | 10.7 | 8.9 | 27.4 | 33.7 (12.3) | 6.8 (2.5) | 5.7 (2.1) | 17.4 (6.4) |
| M5 L | 115.6 (42.2) | 32.5 | 16.5 | 5.7 | 45.3 | 37.6 (13.7) | 19.1 (6.7) | 6.6 (2.4) | 52.3 (19.1) |

**Comment Referee #3:**

4.4 Shouldn't the M5L (and M4L) compositional difference (K for instance) mentioned in this section?

**Reply to comment:**

*We only mentioned the difference in Ti/Al for M5L and M4L, having a lower slope than M1U to M4U. For K the differences are not that obvious for M4L. Indeed, M5L shows differences, which we added in a clause:*

**Changes made in manuscript:**

Page 16, line 22 – 24

"However, spatial differences from east to west indicate the continuous enrichment of especially Al and Fe from M1 to M5, and to an offset of higher K at M5, while Ti stays constant."

**Comment Referee #3:**

section 5, page 5, lines 14-16: It seems as if you assume that dust collected at Iwik derives exclusively from local sources (during emission processes); couldn't distal sources also contribute (during deposition processes)? This should be discussed I think, especially as Iwik might be on the path of large outbreaks involving numerous source regions in West Africa.

**Reply to comment:**

*In this manuscript, the main purpose of the Iwik samples was to compare the lithogenic elements with the element composition in the traps. This turned out to fit nicely, especially for site M1, which is closest to the African coast. We agree that the dust in the Iwik samples might be from distal sources as well. We looked at the backward trajectories which show different wind directions in summer and winter. The average accumulated dust, however, represents the dust we find back in the traps. For a detailed source identification, we propose to refer to Friese et al. (2017).*

**Changes made in manuscript:**

We added the direction of the backward trajectories to the discussion:

Page 19, line 10 – 15, page, line 1

"The spring high can be related to the trade-wind intensities at lower altitudes, whereas the summer high point to sporadic dust storms, invisible on land by satellites. Backward trajectories reveal that the location in Iwik is a transit area for long-range transported Saharan dust (Friese et al. 2017). Overall, the horizontal Saharan dust flux from the land-based MWAC samplers in Iwik cannot be compared directly to the downward flux in residual mass to in the sediment traps. However, they are compositionally the same, especially at the ocean site M1 as indicated by their similar Ti-Al slopes (Fig. 7)."

**Comment Referee #3:**

Page, line 19: The fact there is a downwind decrease in the residual mass flux (and marine biogenic matter fluxes) is interesting but these data are not given in the text or in the table (unless I missed them), as I already pointed out above (4.2.1 to 4.2.5); so please provide numbers (for residual mass fluxes especially, but for the other components as well), especially as such decreases are clearly not obvious from the plots! (cf. fig. 4)

**Reply to comment:**

*We added a table giving the annual fluxes of all components (Table 3). In addition, the entire data set will be available online on pangaea.de.*

**Comment Referee #3:**

page 16, line 3: suggests

**Reply to comment:**

*We corrected the typo.*

**Changes made in manuscript:**

Page 20, line 13

"Overall, we observe a downwind increase in the Al and Fe content of the residual mass fraction, which suggests an increase in clay minerals relative to quartz."

**Comment Referee #3:**

page 16, 5.1, line 17: Is it realistic that the event shown on fig. 10 (i.e., from 31 July) could be sampled in the 18 July- 3 August cups, considering the fact that it takes at least a week for the dust to reach the upper traps (as indicated page 18, lines 1-2)? Especially as dust travels at higher atmospheric levels at that time of the year, as you rightly point it out, and so that it may take longer for the dust to settle through the atmospheric column in the first place.

**Reply to comment:**

*The reviewer is right, cup 19 at M1 would be probably more realistic. And indeed, cup 18 and 19 show higher fluxes in the residual mass fraction. Therefore, it does not change the interpretation as such. In addition to that, most of the dust events last for several days. The shown images show the days where the dust is best seen.*

**Changes made in manuscript:**

We added cup #19 to the text:

Page 20, line 27 – 29

"Such enhanced fluxes in residual mass are found in intervals #16, #18 and #19 at site M1U, and M4U and #16 of M3L, corresponding to the period from 16 June to 2 July, from 18 July to 3 August, and 3 August to 19 August 2013, respectively."

**Comment Referee #3:**

Section 5.1: Overall (for the reason just stated) I find this section should come after section 5.2, since indications on the settling speed in the water column is obviously essential to make the connection between satellite data and water column measurements (as this is actually acknowledged lines 32-33).

**Reply to comment:**

*The current structure, satellite comparison before the downward transport, is from atmosphere to the ocean and is followed by the comparison of other deep ocean mass fluxes off northwest Africa. Therefore, we decided to keep the order of the sections as it is now.*

**No changes made in manuscript:**

**Comment Referee #3:**

page 16, 5.1, lines 22-24: The point you want to make in this sentence (starting by "The diffuse. . .") is unclear to me. Please clarify and a reference should be provided.

**Reply to comment:**

*The diffuse dust cloud is referring to the wide-spread dust in the atmosphere in summer which is visible on satellite imagery. As satellite imagery is nadir, it integrates the amount of dust that is suspended across the whole atmosphere. For this reason, dust that is dispersed very thinly across a wide range of altitudes can be interpreted as high AOD, whereas a thin and dense layer could be interpreted as relatively low AOD.*

**No changes made in manuscript**

**Comment Referee #3:**

page 17, section 5.2, lines 7-8: This is an important point: as you rightly indicate, a significant part of the dust deposited in the ocean needs indeed to be incorporated with biogenic matter (OM in particular) in order to settle in the water column; for this reason, it would have been interesting to plot the residual flux vs the OM flux for all traps as this may provide useful clues on the interplay between the two in the particle settling processes in the different sectors of the Atlantic ocean (see for instance Deuser et al., 1983; Jickells et al., 1990; Bory et Newton, 2000. . . etc.).

**Reply to comment:**

*We tried to avoid plotting the residual mass fraction vs the OM since one depends partially on the other, i.e. the residual mass fraction is the remaining part of the total mass after subtraction of the calcium carbonate, organic matter and biogenic silica. Therefore, it is always a closed sum, promoting artificial negative correlations. Also, given that the composition of the total mass is much less variable than its flux, regression of one constituent mass flux to the other, will produce positive correlations. Bory and Newton (2000) and Jickells et al. (1990), for example, used a different approach to quantify their lithogenic fraction in as much as they analysed the Al content independently and calculated the lithogenic fraction from that. This allows them to correlate the data to the organic matter. However, due to the reviewer's curiosity, we provide the required plot and a table giving the linear regressions and the correlation coefficient $R^2$ below.*

*We did not incorporate the figure in the manuscript.*

[Figure]

| Trap | Linear regression | R2 |
|------|-------------------|-----|
| M1 U | 1.50x + 24.80 | 0.25 |
| M2 U | 1.89x + 2.17 | 0.61 |
| M2 L | 3.27x + 2.41 | 0.57 |
| M3 L | 2.86x + 6.16 | 0.73 |
| M4 U | 1.35x + 9.58 | 0.89 |
| M4 L | 1.12x + 11.01 | 0.58 |
| M5 L | 8.49x – 3.38 | 0.86 |

**Comment Referee #3:**

page 18, section 5.3, first paragraph: As above, the discussion would benefit from the comparison with all existing data (comments on 4.2.1 to 4.2.5).

**Reply to comment:**

*The yearly data of all fluxes are given in table 3 (page 11) and is discussed in sections 4.2.1 to 4.2.5. In addition, we added the following lines to the discussion:*

**Changes made in manuscript:**

Page 22, line 23 – 30

"However, biogenic mass fluxes are about the same as found at M1, where no upwelling-stimulated productivity occurs. At site M1 all biogenic particle fluxes are highest, also in comparison to the other sampling sites. The only exception is the $BSiO_2$ at site M5, which is by far the highest contribution of $BSiO_2$. The low flux of biogenic particles in the mid-Atlantic Ocean (M2 to M4) reflect the limited availability of nutrients and low productivity in the oligotrophic ocean. The higher biogenic mass fluxes closest to either continent (M1 & M5)

may have been enhanced by the higher lithogenic input, especially from higher Saharan dust input at site M1, as mineral dust enhances the settling of organic matter through the water column (Ittekkot et al., 1992; Hamm, 2002).

**Comment Referee #3:**

page 18, section 5.3, lines 15-17: Additional data on ocean productivity (and its seasonal patterns) in the two regions should be provided in order to support such a statement; I would develop the discussion in order to strengthen this hypothesis, or remove this statement, which is too speculative as such.

**Reply to comment:**

*With the addition above, dealing with the same issue, we changed the sentence:*

**Changes made in manuscript:**

Page 22, line 27 – 30

"The higher biogenic mass fluxes closest to either continent (M1 & M5) may have been enhanced by the higher lithogenic input, especially from higher Saharan dust input at site M1, as mineral dust enhances the settling of organic matter through the water column (Ittekkot et al. 1992; Hamm, 2002)."

**Comment Referee #3:**

page 18, section 5.3, lines 28-31: I am not sure I follow the reasoning here: the contribution of an additional "fluvial" (and therefore fine) component on the western side of the transect seems perfectly consistent to me with westward decreasing trend in the grain size. A major fluvial contribution would also be fully consistent with the negative correlation between M1 and M5. So the statement that your grain size data implies that the residual fraction is dominated at all stations by Saharan dust seems to be at odd with the above (and with the last paragraph of page 18 actually).

**Reply to comment:**

*We see the reviewer's point. When you would only look at the grain size data with the decreasing trend from east to west, the fine particles at site M5 do fit in this trend perfectly. Therefore, from the grain size data alone, one would not expect to find fluvial input at first place. Now with the additional data on all marine particle fluxes, we see that site M5 is different from all other sites. With the fluvial input from the west, we can explain the high contribution in residual mass flux as well as the seasonal input of biogenic silica. Nevertheless, the Saharan dust does reach site M5 as well and contributes to the residual mass flux to some extent.*

**Changes made in manuscript:**

We changed the following sentence in the manuscript:

Page 23, line 2 – 5

"Nevertheless, the particle-size distributions at M4 and M5 (Van der Does et al. 2016) alone do fit the general pattern of decreasing particle size with increasing distance from the Saharan dust source, not expecting fluvial input from the west."

**Comment Referee #3:**

page 19, conclusions, line 4: "same" source may be little too affirmative and restrictive (replace with ". . .are similar to the dust collected on the African coast"?).

**Reply to comment:**

*We accepted the suggested wording:*

**Changes made in manuscript:**

Page 23, line 16

"We demonstrate that the lithogenic particles collected in the sediment traps are similar to the dust collected on the African coast."

**Comment Referee #3:**

page 19, conclusions, line 10-11: As above, I see no contradiction here.

**Reply to comment:**

*The reviewer is right. We deleted the clause arguing for the contradiction of the fluvial and aeolian source.*

**Comment Referee #3:**

page 19, conclusions, lines 13-14: This is surprising, and therefore interesting, but the apparent correspondence should be considered with caution (see my comments on page 16, 5.1, line 17).

**Reply to comment:**

*We suggest that the coarser particles settle out faster in both the atmosphere and the ocean water column. In summer, when more dust is in the atmosphere, which is also coarser than in winter, it is plausible that there is also a higher dust deposition onto the ocean. Still, the exact backtracking remains difficult due to assumptions on the dust's transport heights and settling speed in the atmosphere and the ocean. Note that we also find a higher residual mass flux in cup #19 (now added), which fits to this observation as well.*

**No changes made in manuscript**

**Comment Referee #3:**

page 19, conclusions, lines 15-17: It is unclear what [changing] "fluxes" [of Saharan mineral dust] you are referring to (and how it would be calculated). Also, is this underestimation discussed in the text?

**Reply to comment:**

*The wording 'changing' is unclear, it is the changing chemical composition.*

*The complications with the calculation are stated in the method sections from which an underestimation of marine biogenic matter and a subsequently overestimation of the lithogenic mass flux is assumed. However, it is now stated more clearly with an additional sentence.*

**Changes made in manuscript:**

We changed the wording in the conclusions.

Page 23, line 27 – 30

"While the temporal and spatial variability in residual mass fluxes corresponds to the changing chemical composition of Saharan mineral dust, they seem to overestimate the net fluxes due to underestimation of marine biogenic matter."

**Changes made in manuscript:**

We added the following sentence to the method section:

Page 7, line 38 & 39

"Therefore, the residual mass fraction is most likely overestimated while the marine biogenic fraction is underestimated."

**Comment Referee #3:**

Finally, I think the paper does not really document dust "deposition" as indicated in the title (this will surely be best documented by the buoys deployed within the DUSTTRAFFIC project). So you may want to reformulate (I would suggest something in the line of "Compositional changes of Saharan across the Atlantic as recorded in the water column downward particle flux").

**Reply to comment:**

*The floating buoys are active dust samplers, meaning that they suck in the dust-laden air from the atmosphere resulting in dust that is something between the transportation and deposition flux since they are only 2 m from the sea surface. However, we changed the title taking this reviewer's suggestion into account.*

**Changes made in manuscript:**

Page 1, line 1 & 2

"Downward particle fluxes of biogenic matter and Saharan dust across the equatorial North Atlantic"